# Offline Multi-agent Reinforcement Learning via Sequential Score Decomposition

**Dan Qiao**[1]  **Wenhao Li**[2]  **Shanchao Yang**[1]  **Hongyuan Zha**[1]  **Baoxiang Wang**[1 3 †]

## Abstract

Offline cooperative multi-agent reinforcement learning (MARL) faces unique challenges due to the distribution shift between online and offline data collection. While online MARL typically converges to a single coordinated joint policy, offline datasets are often mixtures of diverse cooperative behaviors, resulting in highly multimodal joint behavior distributions. In such settings, independent policy regularization often misaligns joint policy constraints and leads to severe distribution shift. To address this, we propose OMSD, which sequentially decomposes the joint behavior policy into individual conditional distributions and leverages diffusion-based generative models to provide modality-coordinated regularization for each agent. Combined with centralized critic guidance, OMSD achieves coordinated exploration within high-value, in-distribution regions, and avoids out-of-distribution joint actions. Experiments across multiple datasets on various continuous control tasks demonstrate that OMSD consistently achieves state-of-the-art performance, especially in challenging multimodal scenarios. Our results highlight the necessity of modality-aware coordination for robust offline MARL.

## 1. Introduction

Multi-Agent Reinforcement Learning (MARL) has achieved remarkable success in complex decision-making scenarios, including games (Berner et al., 2019; Zhang et al., 2021a), AI-driven economic models (Zheng et al., 2022), power systems (Chen et al., 2021), and traffic control (Ma et al., 2024). Yet online MARL often suffers from poor sample efficiency and a pronounced sim-to-real gap, as simulators fail to capture full complexities in the real-world and real-world exploration is risky and costly. These limitations have motivated offline MARL, which learns coordinated policies from fixed datasets without interacting with the environment during training (Yang et al., 2021; Formanek et al., 2024a). In offline MARL, a central challenge is the distribution shift problem, stemming from the disparity between the learned policy and the data collection policy (Pan et al., 2022; Barde et al., 2024). Beyond the challenges seen in single-agent offline RL (Levine et al., 2020; Prudencio et al., 2023), offline MARL must contend with exponentially large joint state-action spaces, as well as the need for high-quality coordination among agents to achieve common goals. All these challenges make effective policy learning in offline settings very difficult.

To address these challenges, existing offline MARL methods mainly fall into two categories. The first category comprises value-based methods that build on Individual-Global-Maximization (IGM) decompositions (Rashid et al., 2018), typically coupled with conservative value estimation to mitigate critic overestimation problems under limited data coverage (Yang et al., 2021; Pan et al., 2022; Wang et al., 2023b). While these approaches alleviate extrapolation and achieve credit assignment under the Centralized-Training-Decentralized-Execution (CTDE) framework (Yang et al., 2020), the individual $\epsilon$-greedy policy of each agent can still lead to the selection of out-of-distribution (OOD) joint actions, which are often of low quality and may not be covered by the datasets (Matsunaga et al., 2023). The second category constrains policies via behavior-regularized updates or generates trajectories with centralized planners and world models (Matsunaga et al., 2023; Barde et al., 2024; Zhu et al., 2024). Although these methods aim to avoid OOD joint action selection through direct policy constraints, they often rely on local independent regularization for each agent. In cases where dataset policies exhibit substantial behavioral diversity, such local constraints can cause misaligned policy updates at the individual agent level, ultimately hindering the coordination required for an effective joint policy. Furthermore, centralized planners introduce additional burdens in practice, as they often entail high inference costs and require opponent modeling, which may be imprecise (Foerster et al., 2018; Yu et al., 2022), to facilitate the translation into decentralized execution strategies for each agent.

---

[1]School of Data Science, The Chinese University of Hong Kong, Shenzhen, China [2]School of Computer Science, Tongji University, Shanghai, China [3]Vector Institute. Correspondence to: Dan Qiao <danqiao@link.cuhk.edu.cn>, Baoxiang Wang <bxwang@cuhk.edu.cn>.

*Proceedings of the 43$^{rd}$ International Conference on Machine Learning*, Seoul, South Korea. PMLR 306, 2026. Copyright 2026 by the author(s).

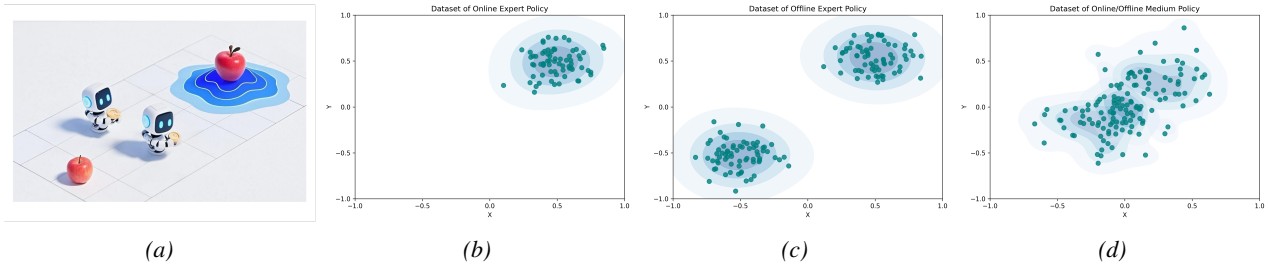

*Figure 1.* (a) Both robots need to cooperatively pick the same one of the two apples in order to receive a reward and end the game. There are two optimal strategies in this game. (b) The online expert policy converges to the optimal joint policy for either single mode due to policy dependence. (c) Offline expert datasets exhibits multimodal optimal joint strategies due to diverse data collection sources. (d) Lower quality datasets demonstrate more pronounced multimodality.

From the perspective of data distribution, the fundamental cause of these limitations lies in the stark difference between online and offline MARL data collection, as exemplified by a simple 2-agent cooperative harvesting task (Fig. 1). This is a common game with multiple Nash Equilibria, where the optimal strategy is for both players to go together to either of the apples. Online MARL resolves this ambiguity via interactive, on-policy adaptation: coupled updates and exploration break symmetry and drive convergence to a single equilibrium, yielding a single-mode joint policy. In contrast, offline MARL datasets are typically mixtures collected from diverse sources with various cooperative policies (Formanek et al., 2023; 2024a), demonstrating highly multimodal behavior. In such scenarios, the multiplicative decomposition of joint policies commonly used in online MARL can lead to biased regularization across agents, as it fails to account for the dependencies introduced by multimodality. Consequently, each agent may be pulled toward different modes, resulting in a misaligned joint policy that lies outside the high-density regions of the dataset.

In this paper, we propose the **Offline MARL with Sequential Score Decomposition** (OMSD) method to achieve coordination regularization under multimodal joint behavior policies. In particular, OMSD sequentially factorizes the joint behavior policy into individual conditional behavior distributions conditioned on both states and prefix-actions, providing an exact chain-rule conditional reference for each agent's Kullback–Leibler (KL) divergence policy constraints. Then the flexible diffusion models are trained to capture complex individual conditional distributions of each agent and estimate the action-space gradient of the KL constraints with score functions (Song et al., 2021). Finally, OMSD combines the individual scores with the centralized critic gradient to guide appropriate exploration within the modality and reduce extrapolation bias with limited data coverage. This design ensures modality-consistent coordination regularization without explicit access to the full joint policy, and guides to high-value in-distribution regions without OOD joint action selection problems. Extensive experiments across various datasets and continuous control

tasks demonstrate that OMSD significantly outperforms existing methods, notably excelling in multimodal scenarios such as medium datasets.

In summary, our contributions are threefold: (i) We identify the multimodal behavior policy distribution introduced by the online-offline data collection gap as the root cause of the difficulty in offline MARL policy coordination, and shed light on how independent regularization can misalign agents and cause the joint action policy distribution to shift; (ii) We develop OMSD, which sequentially decomposes behavior policies and learns diffusion-based conditional scores as a behavior regularizer, which can promote coordinated mode selection without modeling a full joint policy or relying on a planner; (iii) We demonstrate state-of-the-art performance on a multi-agent continuous control task benchmark, effectively handling scenarios with multimodal data distribution. Our source code is available at https://github.com/qiaodan-cuhk/OMSD.

## 2. Preliminaries

### 2.1. Partially Observable Stochastic Game

A partially observable stochastic game (POSG; Hansen et al., 2004) or Markov game is defined as a tuple: $\langle \mathcal{X}, \mathcal{S}, \{\mathcal{A}^i\}_{i=1}^n, \{\mathcal{O}^i\}_{i=1}^n, \mathcal{P}, \mathcal{E}, \{\mathcal{R}^i\}_{i=1}^n \rangle$, where $n$ is the number of agents, $\mathcal{X}$ is the agent space, $\mathcal{S}$ is a finite set of states, $\mathcal{A}^i$ is the action set for agent $i$, $\boldsymbol{\mathcal{A}} = \mathcal{A}^1 \times \mathcal{A}^2 \times \cdots \times \mathcal{A}^n$ is the set of joint actions, $\mathcal{P}(s'|s, \boldsymbol{a})$ is the state transition probability function, $\mathcal{O}^i$ is the observation set for agent $i$, $\boldsymbol{\mathcal{O}} = \mathcal{O}^1 \times \mathcal{O}^2 \times \cdots \times \mathcal{O}^n$ is the set of joint observations, $\mathcal{E}(\boldsymbol{o}|s)$ is the emission function, and $\mathcal{R}^i : \mathcal{S} \times \boldsymbol{\mathcal{A}} \times \mathcal{S} \to \mathbb{R}$ is the reward function for agent $i$. The game progresses over a sequence of stages called the *horizon*, which can be finite or infinite. This paper focuses on the episodic infinite horizon problem, where each agent aims to maximize expected discounted cumulative return.

In a cooperative POSG (Song et al., 2020), the relationship

between agents $x$ and $x'$ is given by:

$$\forall x \in \mathcal{X}, \forall x' \in \mathcal{X} \backslash \{x\}, \forall \pi_x \in \Pi_x, \forall \pi_{x'} \in \Pi_{x'}, \frac{\partial \mathcal{R}^{x'}}{\partial \mathcal{R}^x} \geqslant 0,$$

where $\pi_x$ and $\pi_{x'}$ are policies in the policy spaces $\Pi_x$ and $\Pi_{x'}$, respectively. The inequality condition intuitively means that there is no conflict of interest among any pair of agents. The paper addresses the fully cooperative POSG, also known as the decentralized partially observable Markov decision process (Dec-POMDP; Bernstein et al., 2002), where all agents share the same global reward at each stage, i.e., $\mathcal{R}^1 = \mathcal{R}^2 = \cdots = \mathcal{R}^n$. The optimization goal for Dec-POMDP is defined as: $\max_\Psi \sum_{i=1}^{n} \sum_{t=0}^{\infty} \mathbb{E}_{s_0 \sim p_0, \boldsymbol{o} \sim \mathcal{E}, \boldsymbol{a}_t \sim \boldsymbol{\pi}_\Psi}[\gamma^t r_{t+1}^i]$ where $\Psi := \{\psi^i\}_{i=1}^{n}$ are the parameters of the approximated policies $\pi_{\psi^i}^i : \mathcal{O}^i \to \mathcal{A}^i$, and $\boldsymbol{\pi}_\Psi := \prod_{i=1}^{n} \pi_{\psi^i}^i$ is the joint policy of all agents. Here, $\gamma$ is the discount factor, $p_0$ is the initial state distribution, and $r_{t+1}^i$ is the reward received by agent $i$ at timestep $t+1$ after taking action $a_t^i$ in observation $o_t^i$. In the offline setting, we only have a static dataset of transitions $\mathcal{D} = \{(o_t^m, a_t^m, o_{t+1}^m, r_t^m)\}_{m=1}^{nk}$, where $k$ is the number of transitions for each agent.

## 2.2. Diffusion Probabilistic Models

Diffusion probabilistic models (Sohl-Dickstein et al., 2015; Ho et al., 2020) are a likelihood-based generative framework designed to learn data distributions $q(\boldsymbol{x})$ from offline datasets $\mathcal{D} := \boldsymbol{x}^i$, where $i$ indexes individual samples (Song & Ermon, 2019). A key feature of these models is the representation of the (Stein) score function (Liu et al., 2016), which does not require a tractable partition function.

The model's discrete-time generation procedure involves a forward noising process, defined as $q(\boldsymbol{x}_{k+1}|\boldsymbol{x}_k) := \mathcal{N}(\boldsymbol{x}_{k+1}; \sqrt{\tilde{\alpha}_k}\boldsymbol{x}_k, (1 - \tilde{\alpha}_k)\boldsymbol{I})$, at diffusion timestep $k$. This is paired with a learnable reverse denoising process, $p_\theta(\boldsymbol{x}_{k-1}|\boldsymbol{x}_k) := \mathcal{N}(\boldsymbol{x}_{k-1}|\mu_\theta(\boldsymbol{x}_k, k), \Sigma_k)$, where $\mathcal{N}(\mu, \Sigma)$ represents a Gaussian distribution with mean $\mu$ and variance $\Sigma$. The variance schedule is defined by $\alpha_k \in \mathbb{R}$. Here, $\boldsymbol{x}_0 := \boldsymbol{x}$ is a sample in $\mathcal{D}$, and $\boldsymbol{x}_1, \boldsymbol{x}_2, \ldots, \boldsymbol{x}_{K-1}$ are latent variables, with $\boldsymbol{x}_K \sim \mathcal{N}(\boldsymbol{0}, \boldsymbol{I})$ for appropriately chosen $\tilde{\alpha}_k$ values and a sufficiently large $K$.

Starting with Gaussian noise, samples are iteratively generated through a series of denoising steps. The training of the denoising operator is guided by an optimizable and tractable variational lower bound, with a simplified surrogate loss proposed in (Ho et al., 2020):

$$\mathcal{L}_{\text{denoise}}(\theta) := \mathbb{E}_{k \sim [1,K], \boldsymbol{x}_0 \sim q, \epsilon \sim \mathcal{N}(\boldsymbol{0}, \boldsymbol{I})}\left[\|\epsilon - \epsilon_\theta(\boldsymbol{x}_k, k)\|^2\right] \tag{1}$$

Here, the predicted noise $\epsilon_\theta(\boldsymbol{x}_k, k)$, parameterized by a deep neural network, approximates the noise $\epsilon \sim \mathcal{N}(\boldsymbol{0}, \boldsymbol{I})$ added to the dataset sample $\boldsymbol{x}_0$ to produce the noisy $\boldsymbol{x}_k$ in the noising process.

## 2.3. Policy-Based Offline RL

Policy based methods are successful and widely used in offline RL algorithms. Prior work (Nair et al., 2020) formulates the offline policy optimization problem as:

$$\max_\pi \mathbb{E}_{s \sim \mathcal{D}_\mu}\left[\mathbb{E}_{a \sim \pi(s)}\left[Q_\phi(s, a)\right] - \frac{1}{\beta}\mathcal{D}_{\text{KL}}\left(\pi(\cdot|s)\|\mu(\cdot|s)\right)\right], \tag{2}$$

where $Q_\phi(s, a)$ is a neural network approximation of the state-action value functions $Q^\pi(s, a) := \mathbb{E}_{s_t=s, a_t=a; a_{t+1} \sim \pi}[\sum_{t=0}^{\infty} \gamma^t r(s_t, a_t)]$ under the current policy $\pi$, $\beta$ is a temperature coefficient controlling how far the learned policy deviates from the behavior policy $\mu$. The closed-form solution to this optimization problem (2) is

$$\pi^*(a \mid s) = \frac{1}{Z(s)}\mu(a \mid s)\exp\left(\beta Q_\phi(s, a)\right),$$

where $Z(s)$ is the partition function. A subsequent challenge is to efficiently distill the optimal policy into a parameterized policy $\pi_\theta$. A common approach is minimizing the KL-divergence between $\pi_\theta$ and $\pi^*$ with either forward or reverse direction (Chen et al., 2024). While the optimal policy may be multimodal, meaning it has multiple equivalent policy mode distributions, it is not necessary to express every policy mode explicitly during execution. Therefore, reverse KL is suitable for leveraging its mode-seeking property to capture one feasible mode with a simple parameterized policy, such as a Gaussian or deterministic policy.

**Lemma 2.1** (Behavior-Regularized Policy Optimization (BRPO) (Wu et al., 2019)). *In policy-based offline RL, given an optimal policy $\pi^*$ and a parameterized policy $\pi_\theta$, the policy regularization learning objective with reverse KL-divergence can be written as,*

$$\min_\theta \mathbb{E}_{s \sim \mathcal{D}_\mu} \underbrace{D_{KL}\left[\pi_\theta(\cdot|s)\|\pi^*(\cdot|s)\right]}_{\text{Reverse KL}} \Leftrightarrow \tag{3}$$

$$\max_\theta \underbrace{\mathbb{E}_{s \sim \mathcal{D}_\mu, a \sim \pi_\theta} Q_\phi(s, a) - \frac{1}{\beta}D_{KL}\left(\pi_\theta(\cdot|s)\|\mu(\cdot|s)\right)}_{\text{Behavior-Regularized Policy Optimization}}.$$

## 3. Methodology

### 3.1. Joint Behavior Policy Factorization Mismatch in Offline MARL

The multimodality of joint behavior policy distributions in offline MARL arises from several key factors. First, many cooperative games admit multiple joint policies with similar quality, which is the notorious multiple Nash equilibrium problem. This yields datasets with diverse but equally effective behaviors, complicating policy learning. Second, in large-scale multi-agent systems, especially with homogeneous agents, data collection often anonymizes agent

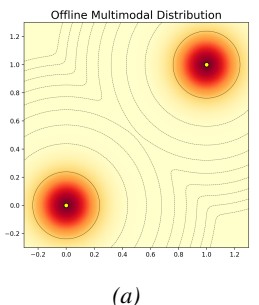 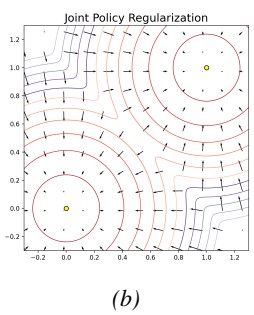 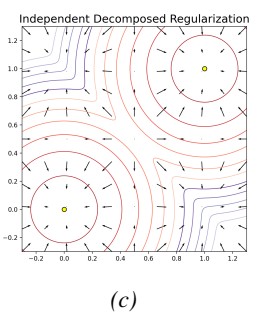 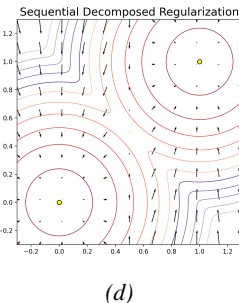

*(a)*       *(b)*       *(c)*       *(d)*

*Figure 2.* From left to right are (a) the original multimodal data distribution; (b) the canonical direction for joint action; (c) the biased direction induced by Combinatorial Mode Shift; (d) the sequential decomposition proposed by OMSD to better align the estimated direction with the canonical direction.

identities (Franzmeyer et al., 2024). Even under a single joint policy, agent trajectories become indistinguishable due to agent interchangeability, introducing inherent symmetry and multimodality. Furthermore, offline datasets are often constructed by mixing demonstrations from various expert and suboptimal strategies due to the high cost of data collection, further increasing behavioral diversity.

Despite this evidence, a common pitfall in offline policy-based methods is the policy factorization assumption, which posits that the joint behavior policy can be factorized as $\mu(\boldsymbol{a}|s) = \prod_{i=1}^{n} \mu_i(a_i|s)$. For example, Al-berDICE (Matsunaga et al., 2023, Eq. 4) implements an occupancy measure penalty using a factorized model $d^D(s, a_i)\pi_{-i}^D(\boldsymbol{a}_{-i}|s, a_i)$, where $-i$ represents all agents except agent $i$, thereby regularizing each agent based on its own marginal behavior and effectively assuming conditional independence. Similarly, DOM2 (Li et al., 2023) trains independent diffusion models for each agent based on local behavioral data, which presupposes that joint behavior can be recovered from marginal distributions. While such factorized regularization is well-motivated and effective in online settings with consistent exploration and joint update adaptation, it will lead to miscoordination of and a significant distribution shift in offline domains where the behavior policy is multimodal and strongly coupled. To formalize this issue, we analyze a stylized scenario and formulate it as a combinatorial mode mixing proposition. The complete proof is deferred to Appendix G.1.

**Proposition 3.1** (**Combinatorial Mode Shift (CMS)**). *Consider a fully cooperative $n$-player game with a single state and continuous action space $\mathcal{A} = [0, 1]^n$. Let $\pi^*$ be the optimal joint policy with two equally weighted optimal modes: $\mathbf{a}_1 = (1, ..., 1)$ and $\mathbf{a}_2 = (0, ..., 0)$. Let $\hat{\pi}$ be a factorized approximation of $\pi^*$ such that $\hat{\pi}(\mathbf{a}) = \prod_{i=1}^{n} \hat{\pi}_i(a_i)$, where each $\hat{\pi}_i$ is learned independently. Then we have each $\hat{\pi}_i$ converges to Uniform$(\{0, 1\})$. The reconstruction of joint policy $\hat{\pi}$ exhibits $2^n$ modes, each with probability $2^{-n}$. The*

*total variation distance between $\pi^*$ and $\hat{\pi}$ is:*

$$\delta_{TV}(\pi^*, \hat{\pi}) = 1 - 2^{1-n}$$

*As $n \to \infty$, $\delta_{TV}(\pi^*, \hat{\pi}) \to 1$, indicating a severe distribution shift.*

This result is intended as a minimal illustration of the CMS pathology rather than a general convergence statement for arbitrary continuous environments. Nevertheless, it reveals a structural failure: even though the expert policy $\pi^*$ is low-entropy and well-coordinated, the factorized approximation $\hat{\pi}$ spreads its support over exponentially many incoherent joint actions. Such a combinatorial mode shift arises because each agent's behavior policy $\mu_i$ is forced to match its own marginal, ignoring inter-agent coordination. Consequently, each agent regresses to an average over modes in its own action space, resulting in an artificial mode mismatch: the high-probability joint actions under $\hat{\pi}$ may not correspond to any trajectory in the dataset.

During offline policy update, marginal behavior regularization can therefore become a misleading constraint: $\mathcal{D}_{\mathrm{KL}}(\pi_{\theta_i}(\cdot|s) \| \mu_i(\cdot|s))$ may steer individual policies toward marginally plausible but jointly incoherent actions. The recovered joint policies lose alignment with any real mode in the datasets, leading to low-efficiency exploration in areas with low data coverage regions. Specifically for the BRPO algorithm, we can summarize the biased regular coordination caused by CMS into combinatorial mode shift.

**Corollary 3.2** (Joint Policy Distribution Shift). *Let $\mu(\mathbf{a}|s)$ be a joint behavior distribution with $K$ coordinated modes over $n$ agents. When each agent regularizes to its own marginal $\mu_i(a_i|s)$ and the joint policy is factorized as $\prod_{i=1}^{n} \pi_i(a_i|s)$, the resulting policy exhibits probability mass on $K^n$ joint actions. As $n$ grows, the total variation distance $\delta_{TV}(\mu, \prod_i \pi_i) \to 1$, indicating a severe distribution shift from the data distribution.*

This pitfall holds whether the underlying BRPO variant is fully independent or uses a centralized critic with the CTDE

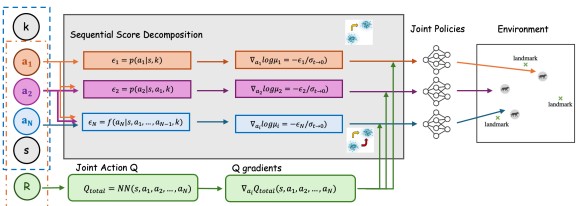

*Figure 3.* Illustration of OMSD: (Top Row) Training sequential diffusion models for each agent to distill score regularization, (Bottom Row) Plugin the sequential score models with joint action Q-gradient.

framework: as long as the regularization is decomposed over agent marginals, policy updates can drift toward spurious high-density configurations unrepresentative of any valid global coordinated behavior in the data. We use a simple two-Gaussian mixture in Fig. 2 to visualize how different policy decomposition choices induce different regularization directions. From an optimization perspective, this failure can be viewed as a structural objective mismatch: marginal regularization constrains the learned policy toward $\prod_i \mu_i(a_i|s)$ rather than the true joint behavior distribution $\mu(\boldsymbol{a}|s)$. Appendix G.2 provides a more detailed discussion of this mismatch and how sequential decomposition removes this factorization error at the behavior-policy level.

### 3.2. Sequential Score Decomposition of Joint Behavior Policy

To address these limitations, we propose a novel policy learning framework named **Offline MARL with Sequential Score Decomposition** (OMSD). Figure 3 summarizes the OMSD workflow, where joint offline data is used to train a global $Q^{tot}(s, \boldsymbol{a})$ and agent-wise conditional diffusion score models. Our method is designed to provide an unbiased reference for coordination regularization and decentralized policy updates in offline learning where joint behavior distributions $\mu(\boldsymbol{a}|s)$ are often complex and highly entangled. Here, "unbiased" refers to using the exact chain-rule decomposition of the behavior policy, rather than independent marginal factorization.

Inspired by coordinate descent and rollout update (Wang et al., 2023a), we address this issue via a *sequential decomposition* of the joint behavior policy. Specifically, we model the behavior distribution as:

$$\boldsymbol{\mu}(\boldsymbol{a}|s) = \Pi_{i=1}^n \hat{\mu}_i(a_i|s, a_{<i}),$$

where $a_{<i}$ denotes the joint actions of all preceding agents, i.e., $a_{<i} = (a_1, \ldots, a_{i-1})$, with each $a_j$ sampled from the corresponding policy $\pi_j(a_j|s)$ for $j = 1, \ldots, i - 1$. This sequential modeling allows each agent to learn its behavior not in isolation but conditionally on earlier agents, capturing inter-agent dependencies without requiring full joint modeling. This structure helps keep individual policy constraints

aligned with the joint behavior distribution, reducing the risk of OOD joint policies.

Following the BRPO framework (Chen et al., 2024), we formulate policy-based offline MARL under the CTDE paradigm. The goal is to learn decentralized policies $\pi_\theta(\boldsymbol{a}|s) = \prod_{j=1}^n \pi_{\theta_j}(a_j|s)$ that maximize the joint value while remaining close to the sequentially decomposed behavior distribution. For the coordinate-wise update of agent $i$, we write:

$$\mathcal{L}^i = \max_{\theta_i} \mathbb{E}_{s \sim \mathcal{D}^\mu, \boldsymbol{a} \sim \pi_\theta(\cdot|s)} Q^{tot}(s, \boldsymbol{a}) \tag{4}$$

$$- \frac{1}{\beta} D_{\mathrm{KL}} \left[ \pi_{\theta_i}(\cdot \mid s) \pi_{\theta_{-i}}(\cdot \mid s) \, \middle\| \, \mu_i(\cdot \mid s, a_{<i}) \mu_{-i}(\cdot \mid s, a_{<i}) \right],$$

where $Q^{tot}(s, \boldsymbol{a})$ represents the joint state-action value estimation, $\mu_{-i}(\boldsymbol{a}_{-i} \mid s, a_{<i}) = \prod_{j<i} \mu_j(a_j \mid s, a_{<j}) \prod_{j>i} \mu_j(a_j \mid s, a_{<j})$ denotes the remaining chain-rule factors of the behavior policy. Equivalently, the behavior reference is the full sequential decomposition $\mu(\boldsymbol{a}|s) = \prod_{j=1}^n \mu_j(a_j|s, a_{<j})$.

When updating $\theta_i$, actions sampled from other agents are treated as fixed and gradients are not propagated through these sampled actions. Moreover, OMSD uses a local coordinate-wise regularization step: the suffix behavior factors in $\mu_{-i}$ are treated as fixed references for this update, so the policy-gradient regularizer for agent $i$ only uses the conditional score $\nabla_{a_i} \log \mu_i(a_i \mid s, a_{<i})$. Under this update rule, we obtain:

$$\nabla_{\theta_i} \mathcal{L}^i = \mathbb{E} \left[ \nabla_{a_i} Q^{tot}(s, \boldsymbol{a}) \big|_{\boldsymbol{a} = \pi_\theta(s)} \tag{5} \right.$$

$$\left. + \frac{1}{\beta} \nabla_{a_i} \log \mu_i(a_i \mid s, a_{<i}) \big|_{a_i = \pi_{\theta_i}(s)} \right] \nabla_{\theta_i} \pi_{\theta_i}(a_i|s),$$

This gradient update allows each agent to balance between maximizing expected return and adhering to its own conditional behavior policy, conditioned on the updated actions of its prefix agents. Such bottom-up sequential guidance helps reduce distributional shift by encouraging later agents to choose actions compatible with the prefix context. The sequential conditioning is used during policy optimization, while execution remains decentralized.

### 3.3. Practical Algorithm

The policy update gradient in Eq. (5) consists of a centralized Q-gradient and a gradient of an unknown logarithm probability distribution. We first adopt centralized offline IQL to learn a joint value function, and then pretrain a conditional diffusion model for each agent, where $-\hat{\epsilon}_i/\sigma_t$ approximates the conditional score $\nabla_{a_i} \log \mu_i(a_i|s, a_{<i})$ at low noise levels. Each agent's score model is trained using only the dataset, and the pretraining is fully parallelizable across agents, making this stage amenable to larger agent teams when parallel resources are available.

Inspired by SRPO (Chen et al., 2024), instead of explicitly modeling the behavior policy distribution $\mu_i(a_i|s, a_{<i})$, we can distill approximations to agent-wise score functions $\nabla_{a_i} \log \mu_i(a_i|s, a_{<i})$ from pretrained diffusion models as gradient regularization into policy update at low noise levels ($t \to 0$), efficiently providing score approximations without requiring sampling actions through a costly denoising process. This transforms policy decomposition into direction-aware regularization, effectively controlling update deviation and encouraging high-value yet conservative exploration. Formally, based on the regularized objective in (4), the practical policy gradient becomes:

$$\nabla_{\theta_i} \mathcal{L}_{OMSD}^i(\theta_i) = \mathbb{E}[\nabla_{a_i} Q_\phi(s, \boldsymbol{a}) \qquad (6)$$

$$+ \frac{1}{\beta} \underbrace{\nabla_{a_i} \log \mu_i(a_i \mid s, a_{<i})|_{a_i = \pi_{\theta_i}(s), a_{<i} = a_{<i}^{\text{new}}}}_{\approx -\hat{\epsilon}_i^*(a_i|s, a_{<i}^{\text{new}}, t)/\sigma_t, \ t \to 0} ] \nabla_{\theta_i} \pi_{\theta_i}(s).$$

To compute the regularization score $\nabla_{a_i} \log \mu_i(a_i|s, a_{<i})$ for $\pi_i^t$, OMSD adopts a sequential update scheme during policy update, where agent $i$ conditions on prefix actions $a_{<i}^{\text{new}}$ sampled from the most recently updated policies $\{\pi_j^{\text{new}}\}_{j<i}$ within the same iteration. Here, $a_{<i}^{\text{new}}$ indicates that, for each agent $i$, the prefix actions are generated by the current versions of agents 1 to $i-1$ after their latest updates in this round. This sequential conditioning is only applied during the policy optimization process to enable coordinated learning, while all agents can still act concurrently and independently during execution. This mechanism encourages the score regularization directions to remain compatible with in-distribution modes of the dataset. To reduce variance in these prefixes and stabilize score estimation, we use deterministic DiLac policies, which preserve expressiveness while avoiding noise amplification in continuous control tasks. Note that the sequential structure is only required during policy update, which provides flexibility for concurrent decentralized execution and parallel diffusion models pretraining. The pseudo code is available in Appendix B. For more details, refer to Appendix H.

## 4. Experiments and Results

In this section, we evaluate the proposed method OMSD on a bandit example and the challenging high-dimensional continuous control multi-agent testbeds (MPE) (Lowe et al., 2017) and MaMuJoCo (Peng et al., 2021). We aim to address the following questions: (i) Can OMSD learn high-quality coordinated policies from sub-optimal datasets with multimodality distribution? (ii) How do policy factorization methods, e.g., Independent Factorization and Sequential Score Decomposition, influence the policy update? (iii) Can OMSD effectively avoid OOD distribution shift problems?

**Environments.** In the bandit example, we design a 2-agent fully cooperative task where the reward function

*Table 1.* Evaluation rewards after convergence for the toy example.

| BRPO-IND | BRPO-JAL | BRPO-CTDE | OMSD (Ours) |
|----------|----------|-----------|-------------|
| 0±1 | 1±0 | 0±1 | 1±0 |

is $r_i = a_1 * a_2$ for $i = 1, 2$. The optimal rewards are achieved with joint actions $[-1, -1]$ and $[1, 1]$. MPE include 3 tasks requiring agents cooperation to conver landmarks or catch the pretrained prey opponent in a 2D environment. In MaMuJoCo, each part of a robot is modeled as an independent agent and learn optimal motions through cooperating with each other. See Appendix D for further details.

**Datasets.** For bandit problem, we generate an action dataset by randomly sampling 1,000,000 times from a 2-Gaussian mixed model with mean values $\mu_0 = [0.8, 0.8], \mu_1 = [-0.8, -0.8]$ and variance $\sigma_0 = \sigma_1 = 0.3$. Considering the inconsistencies in datasets and baselines in previous research, as noted by (Formanek et al., 2024b), we select three of the most well-evaluated benchmarks, the MPE datasets provided by OMAR (Pan et al., 2022), and two MaMuJoCo datasets provided by OG-MARL (Formanek et al., 2023) and OMIGA (Wang et al., 2023c). Each dataset contains datasets of various qualities, ranging from expert to random.

**Baselines.** In the bandit setting, to clearly compare the learning dynamics of different policy decomposition under multimodal datasets, we extend the standard BRPO algorithm to a multi-agent version, including BRPO-JAL (joint action learning), BRPO-IND (independent learning), and BRPO-CTDE. Detailed algorithmic descriptions are provided in the Appendix G. For high-dimensional tasks, we benchmark against state-of-the-art offline MARL methods, including independent learning approaches (BC, MATD3+BC, MA-ICQ, OMAR (Pan et al., 2022)), CTDE value decomposition methods (MA-CQL (Jiang & Lu, 2021) and CFCQL (Shao et al., 2023)), and diffusion-based techniques (MADiff (Zhu et al., 2024) and DoF (Li et al., 2025)).

### 4.1. Bandit Examples

As shown in Table 1, OMSD demonstrates performance comparable to joint action learning algorithm BRPO-JAL, outperforming independent learning and naive CTDE methods with the factorization assumption. Clearly, both BRPO-IND and BRPO-CTDE struggle with OOD joint actions like $[1, -1]$ and $[-1, 1]$. This issue is more pronounced in continuous tasks compared to discrete XOR Matrix Games in (Matsunaga et al., 2023), where behavior policies with limited expressivity often struggle to capture complex multimodal distributions (Wang et al., 2023c).

Furthermore, in Fig. 2, we visualize the policy regularization gradient directions during training by sampling joint actions. Independent factorization methods such as BRPO-

*Table 2.* The average normalized score on offline MARL tasks with OMAR datasets. Shaded columns represent our method. Results are reported as mean ± standard deviation over 5 seeds. Bold and underline denote the best and second-best mean respectively.

| Testbed | Task | Dataset | BC | MA-ICQ | MA-CQL | MA-TD3+BC | OMAR | CFCQL | MADiff-D | DoF-P | OMSD |
|---|---|---|---|---|---|---|---|---|---|---|---|
| MPE | Cooperative Navigation | Expert | 35.0 ± 2.6 | 104.0 ± 3.4 | 98.2 ± 5.2 | 108.3 ± 3.3 | 114.9 ± 2.6 | 112 ± 4.0 | 95.0 ± 11.9 | **126.3 ± 3.1** | 102.3 ± 3.1 (-22.1%) |
| | | Medium | 31.6 ± 4.8 | 29.3 ± 5.5 | 34.1 ± 7.2 | 29.3 ± 4.8 | 47.9 ± 18.9 | 65.0 ± 10.2 | 64.9 ± 17.2 | 60.5 ± 8.5 | **70.1 ± 3.1** (+7.8%) |
| | | Random | -0.5 ± 3.2 | 6.3 ± 3.5 | 24.0 ± 9.8 | 9.8 ± 4.9 | 34.3 ± 5.3 | 62.2 ± 8.1 | 6.9 ± 6.9 | 34.5 ± 5.4 | **69.8 ± 10.3** (+12.1%) |
| | Predator Prey | Expert | 40.0 ± 9.6 | 113.0 ± 14.4 | 93.9 ± 14.0 | 115.2 ± 12.5 | 116.2 ± 19.8 | 118.2 ± 13.1 | 120.9 ± 32.6 | 120.1 ± 6.3 | **161.4 ± 9.4** (+33.5%) |
| | | Medium | 22.5 ± 1.8 | 63.3 ± 20.0 | 61.7 ± 23.1 | 65.1 ± 29.5 | 66.7 ± 23.2 | 68.5 ± 21.8 | 77.2 ± 23.3 | 83.9 ± 9.6 | **137.1 ± 14.1** (+63.0%) |
| | | Random | 1.2 ± 0.8 | 2.2 ± 2.6 | 5.0 ± 8.2 | 5.7 ± 3.5 | 11.1 ± 2.8 | 78.5 ± 15.6 | 3.2 ± 8.9 | 14.8 ± 3.2 | **133.9 ± 16.5** (+70.6%) |
| | World | Expert | 33.0 ± 9.9 | 109.5 ± 22.8 | 71.9 ± 28.1 | 110.3 ± 21.3 | 110.4 ± 25.7 | 119.7 ± 26.4 | 122.6 ± 32.2 | 138.4 ± 20.1 | **163.9 ± 24.1** (+18.4%) |
| | | Medium | 25.3 ± 2.0 | 71.9 ± 20.0 | 58.6 ± 11.2 | 73.4 ± 9.3 | 74.6 ± 11.5 | 93.8 ± 31.8 | 123.5 ± 10.1 | 86.4 ± 10.6 | **160.3 ± 9.2** (+29.8%) |
| | | Random | -2.4 ± 0.5 | 1.0 ± 3.2 | 0.6 ± 2.0 | 2.8 ± 5.5 | 5.9 ± 5.2 | 68 ± 20.8 | 2.0 ± 6.7 | 15.1 ± 3.0 | **141.1 ± 13.0** (+107.5%) |
| | Average Score | | 20.6 ± 3.9 | 55.6 ± 10.6 | 49.8 ± 12.1 | 57.8 ± 10.5 | 64.7 ± 12.8 | 87.3 ± 16.9 | 68.5 ± 16.5 | 75.6 ± 7.8 | **126.7 ± 11.4** (+33.2%) |
| MaMuJoCo (210) | 2-HalfCheetah | Good | 6846 ± 574 | - | - | 7025 ± 439 | 1434 ± 1903 | - | 8246 ± 765 | - | **8619 ± 418** (+4.5%) |
| | | Medium | 1627 ± 187 | - | - | 2561 ± 82 | 1892 ± 220 | - | 2207 ± 51 | - | **2660 ± 125** (+3.9%) |
| | | Poor | 465 ± 59 | - | - | 736 ± 72 | 384 ± 420 | - | 759 ± 40 | - | **866 ± 78** (+14.1%) |
| | 2-Ant | Good | 2697 ± 267 | - | - | 2922 ± 194 | 464 ± 469 | - | **2946 ± 172** | - | 2714 ± 555 (-7.9%) |
| | | Medium | 1145 ± 126 | - | - | 744 ± 283 | 799 ± 186 | - | 1211 ± 154 | - | **1372 ± 107** (+13.1%) |
| | | Poor | 954 ± 80 | - | - | **1256 ± 122** | 857 ± 73 | - | 946 ± 148 | - | 1213 ± 212 (-3.5%) |
| | 4-Ant | Good | 2802 ± 133 | - | - | 2628 ± 971 | 344 ± 631 | - | **3080 ± 85** | - | 2844 ± 152 (-7.7%) |
| | | Medium | 1617 ± 153 | - | - | 1843 ± 494 | 929 ± 349 | - | 1649 ± 224 | - | **1942 ± 293** (+5.3%) |
| | | Poor | 1033 ± 122 | - | - | 1075 ± 96 | 518 ± 112 | - | 1295 ± 127 | - | **1477 ± 192** (+14.1%) |

IND and BRPO-CTDE exhibit miscoordination among independent regularization, potentially leading to OOD joint actions. Benefiting from sequential conditional score decomposition and centralized critics, OMSD better aligns reward-seeking updates with behavior regularization directions, thereby encouraging convergence toward high-value modes within the dataset distribution. Our results highlight OMSD's effectiveness in enforcing the policy update within the joint behavior policy distribution and improving coordination. More detailed discussion about BRPO-IND and BRPO-CTDE can be found in Appendix G.

### 4.2. High-Dimensional Continuous Control Tasks

We further evaluate OMSD on more complex continuous-control tasks in the MPE and MaMuJoCo suites. Table 2 reports results on the OMAR and OG-MARL benchmarks, including normalized scores on MPE and raw episode returns on MaMuJoCo, while Table 3 reports additional MaMuJoCo results on the OMIGA datasets. Most baseline results are taken from the corresponding prior papers and standardized benchmark reports, including CFCQL, MADiff, OMIGA, and DoF. Since MADiff-D reports standard errors, we convert its uncertainty values to standard deviations over five seeds for consistency with OMSD. For MPE, normalized scores are computed as $100 \times (S - S_{Random})/(S_{Expert} - S_{Random})$ following Pan et al. (2022). The expert and random scores for Cooperative Navigation, Predator Prey, and World are $\{516.8, 159.8\}$, $\{185.6, -4.1\}$, and $\{79.5, -6.8\}$, respectively.

OMSD outperforms existing methods on most tasks, with particularly large gains on medium and random datasets where multimodal behavior distributions are more pronounced. On these datasets, OMSD often approaches the high-return episodes observed in the offline datasets (Appendix E), suggesting that sequential score regularization helps identify higher-quality behavior modes while remain-

ing close to dataset-supported regions. On the few tasks where OMSD performs worse than the strongest baseline, we find that policy improvement is mainly limited by the pretrained centralized critic: although the diffusion model can capture multimodal behavior structure, a weak critic provides insufficient reward-improvement guidance. Additional hyperparameter and pretraining details are provided in Appendix D.

To further compare OMSD with diffusion-based offline MARL approaches, we include MADiff-D (Zhu et al., 2024), a decentralized execution variant that uses diffusion models for trajectory planning, and DoF-P (Li et al., 2025), which uses a diffusion actor with factorized noise. We focus on MADiff-D rather than MADiff-C in the main comparison because OMSD targets decentralized execution, while MADiff-C is a centralized generation-based planner. Although MADiff-C can avoid independent-factorization CMS through centralized modeling, its generation-based planning paradigm can be sensitive to compounding sampling errors on low-quality datasets (Zhu et al., 2024). In contrast, OMSD uses diffusion models only as conditional score estimators for policy regularization, avoiding iterative diffusion sampling during execution. Across these diffusion-based baselines, OMSD achieves stronger performance on most tasks, especially in scenarios requiring coordinated behavior. We attribute this advantage to sequential conditional score regularization, which captures inter-agent dependencies and directly shapes policy-gradient directions through behavior-aware regularization.

To further evaluate generality across dataset sources and larger-agent settings, we report additional MaMuJoCo results trained on the OMIGA datasets in Table 3. OMSD achieves an average improvement of 73.9% over the strongest reported baseline, with especially large gains on mixed datasets such as Medium-Expert and Medium-Replay. These results further support the benefit of modeling multi-

*Table 3.* Experiment results on the `MaMuJoCo` environments with OMIGA ([Wang et al., 2023b](#)) datasets.

| Task | Dataset | BCQMA | CQLMA | ICQ | OMAR | OMIGA | OMSD (ours) |
|---|---|---|---|---|---|---|---|
| 6-HalfCheetah | Expert | 2992.71±629.65 | 1189.54±1034.49 | 2955.94±459.19 | -206.73±161.12 | 3383.61±552.67 | **5545±156** (+64%) |
| | Medium-Expert | 3543.70±780.89 | 1194.23±1081.06 | 2833.99±420.32 | -253.84±63.94 | 2948.46±518.89 | **5237±46** (+48%) |
| | Medium-Replay | -333.64±152.06 | 1998.67±693.92 | 1922.42±612.87 | -235.42±154.89 | 2504.70±83.47 | **4582±52** (+83%) |
| | Medium | 2590.47±1110.35 | 1011.35±1016.94 | 2549.27±96.34 | -265.68±146.98 | 3608.13±237.37 | **4695±62** (+30%) |
| 3-Hopper | Expert | 77.85±58.04 | 159.14± 313.83 | 754.74± 806.28 | 2.36± 1.46 | 859.63±709.47 | **3595 ± 66** (+329%) |
| | Medium-Expert | 54.31±23.66 | 64.82±123.31 | 355.44±373.86 | 1.44±0.86 | 709.00±595.66 | **3568 ± 45** (+403%) |
| | Medium | 44.58±20.62 | 401.27±199.88 | 501.79±14.03 | 21.34±24.90 | 1189.26± 544.30 | **3360 ± 276** (+183%) |
| 2-Ant | Expert | 1317.73±286.28 | 1042.39±2021.65 | 2050.00±11.86 | 312.54±297.48 | 2055.46±1.58 | **2191 ± 46** (+6.6%) |
| | Medium-Expert | 1020.89±242.74 | 800.22±1621.52 | 1590.18±85.61 | -2992.80± 6.95 | 1720.33±110.63 | **2002 ± 124** (+16.4%) |
| | Medium-Replay | 950.77±48.76 | 234.62±1618.28 | 1016.68±53.51 | -2014.20±844.68 | **1105.13±88.87** | 1009 ± 43(-8.7%) |
| | Medium | 1059.60±91.22 | 533.90±1766.42 | 1412.41±10.93 | -1710.04±1588.98 | 1418.44±5.36 | **1619 ± 77** (+14.2%) |
| | Average | 1210.82±313.12 | 784.56±1044.66 | 1631.17±267.71 | -667.37±299.29 | 1954.74±313.48 | **3400±90** (+73.9%) |

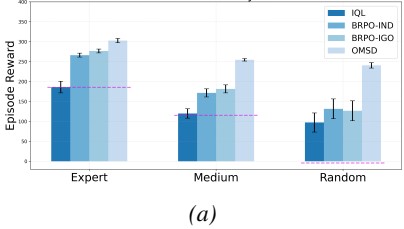
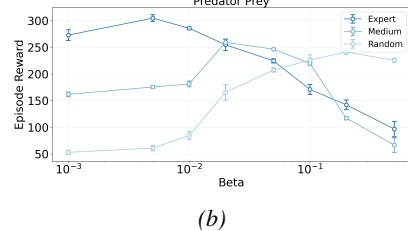
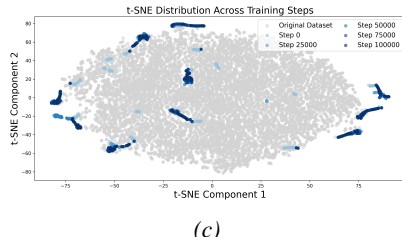

|         (a)         |         (b)         |         (c)         |

*Figure 4.* (a) Comparison of pre-trained IQL and post-trained algorithms. (b) Regularization term $\beta$ for OMSD performance. (c) t-SNE Visualization of policy evolution during OMSD training.

modal joint behavior distributions with coordinated conditional score regularization. From a scalability perspective, OMSD decouples score-model pretraining across agents: each conditional score model can be trained independently once the offline dataset is fixed. The sequential dependency is only used during policy optimization through prefix actions, and does not impose sequential execution at test time. This makes OMSD amenable to larger-agent continuous-control settings, as reflected by the 6-agent HalfCheetah experiments. Additional `MaMuJoCo` results are provided in Appendix C.

### 4.3. Ablation Study

**Does Score Decomposition Method Matter?** To investigate the impact of our proposed sequential score decomposition mechanism, we conduct a series of ablation studies. For a fair comparison, we compare OMSD against BRPO-IND and BRPO-CTDE as described in Sec. F. As shown in Fig. 4a, OMSD consistently outperforms both the pretrained IQL and factorization methods, as well as the overall dataset quality. The average episode reward across datasets is indicated by a purple dashed line. The notable improvement over the pretrained IQL highlights OMSD's ability to effectively combine global critic signals with policy constraints, enabling more reliable offline policy improvement. In contrast, the performance gap between OMSD and BRPO-CTDE illustrates that inappropriate score decomposition can lead to poorly coordinated joint policies that suffer from OOD

actions, ultimately degrading overall performance. The dotted lines in the figure indicate the average and maximized absolute return of the training datasets.

To demonstrate our method's insensitivity to update order, we conducted randomized ordering experiments on the OMIGA Hopper-v2 datasets. Experimental results show that, with the same pre-training parameters and OMSD training parameter settings, changing the update order does not significantly impact performance, suggesting that the sequential order mainly serves as a training-time coordination mechanism rather than a strong inductive bias. Additional results are provided in the Appendix D.7.1 and D.7.4. We further compare diffusion score estimators with simpler conditional density estimators, including GMMs and normalizing flows, in Appendix D.7.5.

**Hyperparameters.** Since policy-based offline methods are sensitive to the degree of behavior regularization, we conduct a systematic study on the influence of the regularization coefficient $\beta$ as shown in Fig. 4b. Specifically, we sweep $\beta$ over the set $\{0.001, 0.005, 0.01, 0.02, 0.05, 0.1, 0.3, 0.5\}$. Our results show that the optimal value of $\beta$ depends strongly on the quality of the dataset. Expert-level datasets often benefit from stronger policy constraints (e.g., $\beta = 0.001$), preserving high-quality behaviors. In contrast, lower-quality datasets such as random favor weaker regularization (e.g., $\beta = 0.3$), allowing the policy to deviate from suboptimal demonstrations and encourage more exploratory behavior. The stable performance over a broad

range of $\beta$ values also indicates that OMSD is not overly sensitive to the relative weighting between critic guidance and score regularization. For detailed experimental results on additional tasks, please refer to the Appendix D.7.2.

**How does OMSD stay close to dataset-supported modes?** We observe that OMSD achieves strong performance gains on low-quality datasets, where prior methods often struggle. To investigate this, we visualize the learning policy checkpoints via t-SNE (Van der Maaten & Hinton, 2008) by sampling state-action pairs from the policy and comparing them to the dataset distribution. As shown in Fig. 4c, OMSD captures the underlying multimodal structure and concentrates around high-reward regions within the dataset support. This provides qualitative evidence that OMSD can exploit the critic as a reward landmark while remaining close to dataset-supported regions, which enables stable policy improvement.

## 5. Related Works

**Offline MARL.** Early research in offline MARL mainly made efforts to extend the pessimistic principles from offline single-agent RL with independent learning paradigm. For example, MAICQ (Yang et al., 2021) and MABCQ (Jiang & Lu, 2021) extended the pessimistic value estimation such as CQL to multi-agent and discuss the extrapolation error under exponential increasing dimension of joint actions space problem. Furthermore, OMAR (Pan et al., 2022) dealt with the local optima with zero-th order optimization. Motivated by this, CFCQL (Shao et al., 2023) further improved OMAR with counterfactual value estimation to avoid over-pessimistic value estimation. Recently, MACCA (Wang et al., 2025) and OMIGA (Wang et al., 2023b) has incorporated causal credit assignment technique and the IGM principle into the offline value decomposition process to enhance the credit assignment. In SIT (Tian et al., 2023), authors recognized the data-imbalance problem and handle it with reliable credit assignment technique. On the other hand, AlberDICE (Matsunaga et al., 2023) and MOMA-PPO (Barde et al., 2024) recognized and addressed OOD joint action coordination problems with alternative best response and world model based planning. Our method aligns in this direction and try to model complex behavior policies with diffusion models. BRUD (Tilbury et al., 2024) discusses the failure of policy updates caused by different data points under offline MADDPG-style algorithm. The prioritised dataset sampling mechanism is proposed to ensure that the sampled data in the current batch is close to the distribution of the updated policy. Although this paper considers the impact of data points on policy learning under offline MARL, MADDPG-type modeling still ignores the multimodal characteristics of the joint behavior policy distribution. Besides, there are also some works following the

trajectory generation route, such as MAT (Wen et al., 2022), MADT (Meng et al., 2023), and MADTKD (Tseng et al., 2022). These methods are beyond our scope.

**Diffusion Models in RL.** Recently, motivated by the great advantage of diffusion models, RL researchers turn to seek the possibilities of introducing diffusion models into RL area. Previous works can be typically divided into three topics: serving as planner, serving as policy, and serving for data augmentation. Our method mainly falls in the second topic, where simple density estimators can suffer from poor mode coverage under multimodal data distributions. Diff-QL (Wang et al., 2023c) and SfBC (Chen et al., 2022) used diffusion model to represent the behavior policy and generate a batch of candidate actions with diffusion models, then use resampling to choose the executive actions. These methods suffer the inherent drawback of slow inference process of diffusion models. For this reason, some works tried to accelerate the sampling process of diffusion actor. EDP (Kang et al., 2024) and consistency-AC (Ding & Jin, 2024) leveraged the advanced diffusion models to accelerate the action sampling in RL tasks. Diff-DICE (Mao et al., 2024) investigated guiding and selecting paradigm in diffusion-based RL and avoid OOD actions by proposing a guide-then-select mechanism. Recently, there are few works such as MADiff (Zhu et al., 2024) and DoF (Li et al., 2025), which take diffusion models as a centralized planner or actors. DoF (Li et al., 2025) introduces a novel diffusion-based factorization framework that explicitly models multi-agent interactions, representing significant progress in this domain. Similarly, DOM2 (Li et al., 2023) adopts diffusion models as a data augmentation tool to synthesize interaction-aware trajectories, improving cooperative behavior on shifted environments. While these works span diverse methodologies, our approach aligns with efforts to address OOD joint action challenges and complex behavior policies by leveraging advanced diffusion-based mechanisms.

## 6. Conclusion

In this paper, we study the key challenge of multimodal joint behavior policies in offline MARL and propose the sequential score decomposition algorithm OMSD with diffusion models. To our knowledge, OMSD is the first policy decomposition-based offline MARL algorithm explicitly addresses multimodal behavior policies, leveraging the decomposed score functions distilled from diffusion models to regularize the policy update gradients. Experiment results demonstrate the superiority of our methods OMSD and the effectiveness of policy improvement with coordinate action selection. One future work aims to develop more precise and optimal policy decomposition methods to enhance the ability of policy-based offline MARL methods.

# Acknowledgements

Baoxiang Wang and Dan Qiao are partially supported by the National Natural Science Foundation of China (No. 72394361) and the Shenzhen Science and Technology Program (Nos. JCYJ20250604141218024 and JCYJ20250604141032005). Wenhao Li is supported by the NSFC (No. 62406270) and the STCSM Shanghai Rising-Star Program (No. 24YF2748800). The authors would like to express their sincere gratitude to Yue Lin, Han Wang, Ang Li, and Jiawei Xu from the Chinese University of Hong Kong, Shenzhen (CUHKSZ) for their insightful discussions and constant support. We also thank the anonymous reviewers for their constructive comments and suggestions.

# Impact Statement

This work advances offline multi-agent reinforcement learning (MARL) by addressing the challenge of coordination-aware decomposition of multimodal joint action behavior distributions. Our methods improve coordination and decision-making in multi-agent systems, with potential applications in robotics, autonomous vehicles, and collaborative AI systems. By enabling more effective offline learning, our approach reduces the need for risky online exploration in safety-critical domains.

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

# A. Limitation and Future Work

Our current work focuses on continuous control offline MARL tasks, and we have not yet validated OMSD on discrete or hybrid action spaces. OMSD also introduces additional computation for pretraining conditional diffusion score models, although this stage is decoupled across agents and can be parallelized. Our experiments cover up to six-agent continuous-control tasks, leaving the evaluation on larger-scale teams to future work. Finally, OMSD depends on a pretrained centralized critic for reward-improvement directions, so poor critic estimates under very low-quality data may limit policy improvement.

# B. OMSD Pseudo Code

Here we provide the pseudo-code of our algorithm OMSD.

---

**Algorithm 1** OMSD Algorithm

---

1: **Input**: Offline dataset $\mathcal{D}^\mu$
2: **// Critic Pretraining**
3: **for** critic training step **do**
4:     Train centralized joint critic $Q^{tot}$
5: **end for**
6: **// Score Pretraining (Parallelizable)**
7: **for all** agent $i = 1, \ldots, n$ **in parallel do**
8:     Pretrain conditional diffusion score model $\hat{\epsilon}_i$ on $\mathcal{D}^\mu$
9: **end for**
10: **// Policy Optimization (Sequential Update)**
11: **for** policy gradient step **do**
12:     **for** agent $i = 1, \ldots, n$ (in order) **do**
13:         Sample prefix actions $a_{<i}$ using latest policies $\{\pi_j\}_{j<i}$
14:         Update $\theta_i \leftarrow \theta_i + \alpha \nabla_{\theta_i} \mathcal{L}^i_{OMSD}(\theta_i)$ (Eq. 6)
15:     **end for**
16: **end for**

---

# C. Additional Experiments on `MaMuJoCo`

Experimental results on Table 4 are trained on the 2-agent Halfcheetah dataset provided by OMAR (Pan et al., 2022). In this experiment, OMSD achieves the best performance in three scenarios across four experiment settings. The most significant improvement is observed on the Medium-Replay dataset, highlighting the challenge posed by the severe multimodal distribution of joint behavior policies on mixed-quality datasets to offline MARL algorithms, which can be effectively captured and handled by our methods. Poor performance on the random-quality dataset is attributed to the difficulty of learning the centralized critic on this dataset. Furthermore, since the behavioral policies on the poor dataset are the worst, the policy regularization learned by the diffusion model struggles to provide stable policy constraints and performance improvements. This suggests that our approach may benefit from combining it with better critics from more robust value-based offline MARL training methods.

*Table 4.* Experiment results on the `MaMuJoCo` environments with OMAR (Pan et al., 2022) datasets.

| Task | Dataset | MA-ICQ | MA-CQL | MA-TD3+BC | OMAR | CFCQL | OMSD |
|---|---|---|---|---|---|---|---|
| 2-HalfCheetah | Expert | $110.6 \pm 3.3$ | $50.1 \pm 20.1$ | $114.4 \pm 3.8$ | $113.5 \pm 4.3$ | $\underline{118.5 \pm 4.9}$ | **119.0 ± 1.3** (+0.4%) |
| | Medium | $73.6 \pm 5.0$ | $51.5 \pm 26.7$ | $75.5 \pm 3.7$ | $80.4 \pm 10.2$ | $\underline{80.5 \pm 9.6}$ | **81.4 ± 7.2** (+1.2%) |
| | Med-Replay | $35.6 \pm 2.7$ | $37.0 \pm 7.1$ | $27.1 \pm 5.5$ | $57.7 \pm 5.1$ | $\underline{59.5 \pm 8.2}$ | **78.9 ±4.4** (+32.6%) |
| | Random | $7.4 \pm 0.0$ | $5.3 \pm 0.5$ | $7.4 \pm 0.0$ | $13.5 \pm 7.0$ | **39.7±4.0** | $15.6 \pm 4.2$ (-60.7%) |

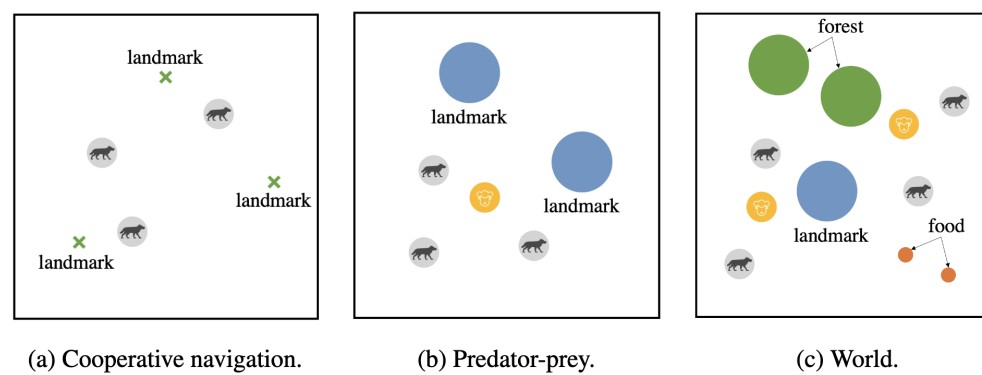

(a) Cooperative navigation.  (b) Predator-prey.  (c) World.

*Figure 5.* `MPE` environments. (Pan et al., 2022)

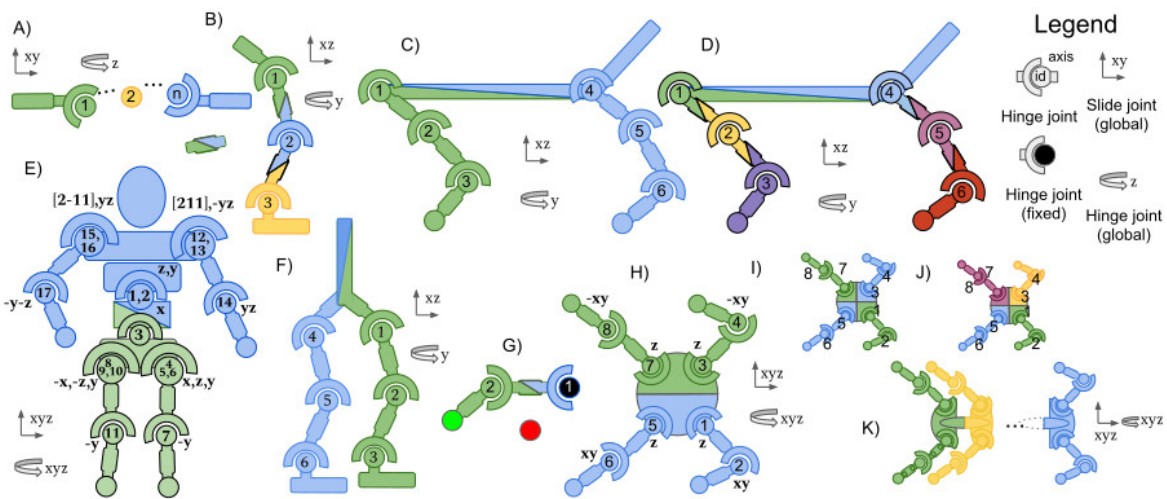

*Figure 6.* `MaMuJoCo` environments. (Peng et al., 2021)

## D. Experimental Details

In this section, we highlight the most important implementation details for the OMSD and baselines. More details can be found in our open-source code.

### D.1. Environment Details

We use the open-source implementations of multi-agent particle environments[1] (Lowe et al., 2017) and `MaMuJoCo`[2] (Peng et al., 2021). Fig. 5 and Fig. 6 illustrate the rendered environments.

In Cooperative Navigation task, 3 learning agents need to cooperatively spread to 3 landmarks, where the common rewards are based on the distances away from landmarks with collusion penalties. In Predator Prey, 3 predators are trained to catch a moving prey, which challenge the predators to surround the prey with high degree of coordination. In world, the original settings involves 4 slower cooperating predators to catch 2 faster preys, where the preys are rewarded by avoiding being captured and eating foods. However, the offline datasets provided by OMAR is trained with 3 slower predators and 1 prey. In 2-agent HalfCheetah task, a halfcheetah with 6 joints need to keep moving forward. The 6 joints are divided into two groups, where each agent controls 3 joints, representing the front legs and the hind legs respectively.

Specifically, we noticed that several commonly used datasets have different settings for `MaMuJoCo`, which affects the dimension of the observation space. Taking the `2-agent Halfcheetah` dataset as an example, the OMAR dataset uses

---

[1]https://github.com/openai/multiagent-particle-envs
[2]https://github.com/schroederdewitt/multiagent_mujoco

`obsk=0`, disregarding neighbor information, resulting in a state space dimension of `state_dim=17` and an observation space dimension of `obs_dim=6`. OMIGA customizes an environment wrapper, and the returned observation variables are actually global state variables, with both the state space and observation space dimensions of `state_dim=17`. The OG-MARL dataset additionally sets `obsk=1` to consider neighbor information and `global categories: {qvel, qpos}`, causing the observation space to be expanded to `obs_dim=13` dimensions. MADiff, due to its transformer structure, adds one-hot encoding to OG-MARL to represent the agent ID, resulting in an observation dimension of `obs_dim=15` dimensions. To ensure fairness, this paper uniformly follows the original dataset collection process settings, removing the one-hot ID from the MADiff dataset to ensure it is independent of agent ID information. Besides, both OMAR[3] and OMIGA[4] employs `mujoco 200` and `mamujoco=0.0.1`, while OG-MARL[5] employs `mujoco 210` and `mamujoco=1.1.0`. The different versions of the mujoco suites will also lead to obvious performance differences.

### D.2. Baseline Settings

In this section, we provide additional details for each of the baseline algorithms. All scores of baselines are derived from the standardized scores reported in the MADiff (Zhu et al., 2024) and the DoF (Li et al., 2025). Consider that OMSD is developed as a CTDE algorithm for continuous control tasks, we select the decentralized version MADiff-D and DoF-P. The open-sourced implementations of baselines are from MADDPG (Iqbal & Sha, 2019)[6], OMAR (Pan et al., 2022)[7], CFCQL (Shao et al., 2023)[8], OMIGA (Wang et al., 2023b)[9], MADiff (Zhu et al., 2024)[10], and DoF (Li et al., 2025)[11].

### D.3. Network Architecture

The hyperparameter and network architecture settings for pre-training primarily follow those of the standard IQL algorithm (Kostrikov et al., 2022) and SRPO algorithm (Chen et al., 2024).

For the centralized critic model, we adapt it from the standard IQL implementation[12]. This model consists of a deterministic policy network, a state-action value network (Q-net) with double-Q learning for stabilized training, and a state value network (V-net). All networks are structured as 2-layer MLPs with 256 hidden units and ReLU activations. The deterministic policy network is optimized using annealing AdamW with a learning rate of $3 \times 10^{-4}$, while the value networks are trained using Adam with a fixed learning rate of $3 \times 10^{-4}$.

The diffusion behavior model is implemented as a 2-layer U-Net with 512 hidden units. The time embedding dimension is set to 64, and the embedding dimension for concatenated input (state and actions) is 32. The learning rate is $3 \times 10^{-4}$.

The policy model is a Dilac policy represented by a 2-layer MLP with 256 hidden units and ReLU activations. It is trained using the Adam optimizer with a learning rate of $3 \times 10^{-4}$ and a batch size of 512. The training process consists of 1.0 million gradient steps for `MaMuJoCo` tasks and 0.1 million gradient steps for `MPE` tasks. The key hyperparameters for OMSD are summarized in Table 5.

### D.4. Pretrain Critic Models

In this section, we provide a detailed explanation of the pre-training process for the critic networks. The network structures and parameter settings are consistent with those described in the previous section. We pre-trained two types of critic networks: independent critic networks and joint action learning critic networks. For the independent critic networks, each agent's input consists of the concatenation of its individual dataset's states and actions, with the network learning each agent's behavior independently. In contrast, the joint action learning critic network adopts a centralized approach, where the input comprises the concatenated joint states (observations) and joint actions of all agents, enabling a global perspective for joint decision-making. All pre-trained critics were trained for 200-500 epochs with checkpoints saved every 50 epochs. In

---

[3]OMAR datasets: `https://github.com/ling-pan/OMAR`
[4]OMIGA datasets: `https://cloud.tsinghua.edu.cn/d/dcf588d659214a28a777/`
[5]OG-MARL datasets: `https://github.com/instadeepai/og-marl`
[6]MADDPG source code: `https://github.com/shariqiqbal2810/maddpg-pytorch`
[7]OMAR source code: `https://github.com/ling-pan/OMAR`
[8]CFCQL source code: `https://github.com/thu-rllab/CFCQL`
[9]OMIGA source code: `https://github.com/ZhengYinan-AIR/OMIGA`
[10]MADiff source code: `https://github.com/zbzhu99/madiff`
[11]DoF source code: `https://github.com/xmu-rl-3dv/DoF`
[12]IQL source code: `https://github.com/ikostrikov/implicit_q_learning`

*Table 5.* Hyper-Parameters for OMSD

| Algorithm | Hyper-Parameter Name | Value |
|---|---|---|
| All | Batch Size | 512 |
| All | Optimizer | Adam |
| All | Learning Rate | $3 \times 10^{-4}$ |
| All | Hidden Activation Function | ReLU |
| All | Discount Factors of RL $\gamma$ | 0.99 |
| All | Soft Update Rate of Target Networks $\tau$ | 0.005 |
| All | MPE Episode Length | 25 |
| All | `MaMuJoCo` Episode Length | 1000 |
| All | Buffer Size | 1e6 |
| All | Reward Scale | 1 |
| Critic & Diffusion Models | Training Epochs | 200 |
| Critic & Diffusion Models | Training Steps in Each Epoch | 10000 |
| Critic & Diffusion Models | Actor Blocks | 2 |
| Critic Models | Q-Network Layers | 2 |
| Diffusion Models | Time Gaussian Projection Dims | 32 |
| Diffusion Models | Time Embedding Dims | 64 |
| Diffusion Models | State-action Embedding Dims | 32 |
| Diffusion Models | Resnet Hidden Dims | 512 |
| Diffusion Models | Dilac Policy Learning Rate | 3e-4 |

subsequent OMSD training, the critic generally loads the checkpoint from the final epoch.

During the optimization process, we made adjustments to various hyperparameters and design choices, uncovering some important insights. First, the temperature and quantile regression coefficient $\tau$ were found to significantly affect the performance of pre-trained IQL. We performed a sweep of $\tau$ values in the range of [0.3, 0.5, 0.7, 0.9] and temperature values in the range of [1, 3, 5, 7, 10] across datasets of different quality and reported the optimal hyperparameters in Tables 6 and 7. Second, regarding the clamping of the advantage function, we initially clamped the exponential advantage term `exp_adv` at a maximum value of 100. However, we later tried directly restricting the advantage values to the range [-1, 1], which improved training stability in certain cases.

However, in the `MPE` environment, we encountered some challenges and issues that significantly impacted OMSD's performance. First, in medium replay datasets compared to those of other quality levels, the training speed was approximately 3 times faster than expected. Additionally, the resulting performance failed to learn meaningful signals. We hypothesize this is due to the sample volume of medium replay datasets being significantly lower than that of others, with medium replay containing only 62,500 samples, whereas datasets of other quality levels contain 1,000,000 samples. The poor performance may be influenced by the dataset's characteristics or overfitting during training, which requires further investigation and resolution. Notably, such issues were not observed in datasets from other environments, such as `MaMuJoCo`.

### D.4.1. MPE TASKS

Since `MPE` tasks consist of only 25 steps per episode, significantly fewer than the 1000 steps per episode in `MaMuJoCo`, we follow the settings of Clean Offline RL (Tarasov et al., 2023) to train IQL algorithms 500 epochs with 1000 update steps per epoch. Below are the hyperparameters for all three `MPE` tasks:

### D.4.2. MaMuJoCo TASKS

The training parameters are aligned with SRPO and have been shown to work effectively. Specifically, for the critic, we use 10,000 steps per epoch for a total of 200 epochs. The quantile regression coefficient $\tau$ is set to 0.9 for maze tasks and 0.7 otherwise, while the temperature $\beta$ is fixed at 10. Additionally, the exponential advantage term "`exp_adv`" is clamped to a maximum value of 100 to ensure training stability.

For the `MaMuJoCo` tasks, the hyperparameters are outlined as follows. Please note that the `2-HalfCheetah` dataset used in OMAR is collected by the mujoco-200 engine, while the datasets in OMIGA, and OG-MARL (Formanek et al., 2023) and

*Table 6.* IQL Training Hyperparameters in MPE

| Environment | Task | Hyper Parameter Name | Value |
|---|---|---|---|
| Global | | Training Steps/Epoch | 1000 |
| | | Epochs | 500 |
| Cooperative Navigation | Expert | temperature | 3.0 |
| | Expert | $\tau$ | 0.5 |
| | Medium | temperature | 0.5 |
| | Medium | $\tau$ | 0.7 |
| | Random | temperature | 0.5 |
| | Random | $\tau$ | 0.5 |
| Predator Prey | Expert | temperature | 7.0 |
| | Expert | $\tau$ | 0.7 |
| | Medium | temperature | 1.0 |
| | Medium | $\tau$ | 0.5 |
| | Random | temperature | 5.0 |
| | Random | $\tau$ | 0.7 |
| World | Expert | temperature | 3.0 |
| | Expert | $\tau$ | 0.5 |
| | Medium | temperature | 1.0 |
| | Medium | $\tau$ | 0.9 |
| | Random | temperature | 7.0 |
| | Random | $\tau$ | 0.7 |

MADiff (Zhu et al., 2024) are collected using mujoco-210. Engine differences may lead to significant differences in training performance. To maintain consistency, the main experiments in this paper all use the latest datasets and a mujoco-210 configuration.

*Table 7.* IQL Training Hyperparameters in `MaMuJoCo`

| Environment | Task | Hyper Parameter Name | Value |
|---|---|---|---|
| Global | | Training Steps/Epoch | 10000 |
| | | Epochs | 200 |
| 2-HalfCheetah | Expert | temperature | 3.0 |
| | Expert | $\tau$ | 0.7 |
| | Medium | temperature | 3.0 |
| | Medium | $\tau$ | 0.7 |
| | Medium-Replay | temperature | 3.0 |
| | Medium-Replay | $\tau$ | 0.7 |
| | Random | temperature | 5.0 |
| | Random | $\tau$ | 0.5 |

### D.5. Pretrain Diffusion Models

For diffusion models, we follow the SRPO (Chen et al., 2024) settings with slight modifications to improve training efficiency. Specifically, we reduce the number of layers from 3 to 2. The noise settings are defined as $t = $ `torch.rand($a$.shape[0], device $= s$.device)` $\times 0.96 + 0.02$. For the base SRPO framework, we use a hidden dimension of 64, a $\tau$ target network soft update rate of 0.01, a learning rate of 0.01, and the Annealing AdamW optimizer. Denoising is performed with 20 steps, while the denoising DDPM model operates with 5 steps using a beta schedule set to the "vp" strategy.

In this study, we pretrained three types of diffusion models: (1) the independent diffusion model, (2) the joint action learning diffusion model, and (3) the sequential diffusion model. In the independent diffusion model, each agent's input consists of a concatenation of its individual dataset's state and action. For the joint action learning diffusion model, learning is treated as a centralized process, with inputs comprising the concatenated joint states (observations) and joint actions of all agents. Finally, the sequential diffusion model extends this idea by incorporating the preceding agents' actions as a prefix to the input. Combined with each agent's own state and action, this adjustment results in task-specific variations in input dimensionality for each agent. The hyperparameters are shown in Tables 8 and Table 9.

### D.5.1. MPE TASKS

Here are the hyperparameters for all three tasks in `MPE` environments shown in Table 8.

*Table 8.* Diffusion Models Training Hyperparameters in MPE

| Environment | Task | Hyper Parameter Name | Value |
|---|---|---|---|
| Global | | Training Steps | 100000 |
| | | Annealing Epochs | 10 |
| Cooperative Navigation | Expert | $\beta$ | 0.001 |
| | Medium | $\beta$ | 0.005 |
| | Random | $\beta$ | 0.05 |
| Predator Prey | Expert | $\beta$ | 0.005 |
| | Medium | $\beta$ | 0.05 |
| | Random | $\beta$ | 0.5 |
| World | Expert | $\beta$ | 0.01 |
| | Medium | $\beta$ | 0.05 |
| | Random | $\beta$ | 0.5 |

### D.5.2. MAMUJOCO TASKS

Here are the hyperparameters for `MaMuJoCo` comes from OMAR (Pan et al., 2022) and MADiff (Zhu et al., 2024) shown in Table 9.

*Table 9.* Diffusion Models Training Hyperparameters in `MaMuJoCo`

| Environment | Task | Hyper Parameter Name | Value |
|---|---|---|---|
| Global | | Training Steps | 100000 |
| | | Annealing Epochs | 10 |
| 2-HalfCheetah 200 | Expert | $\beta$ | 0.001 |
| | Medium | $\beta$ | 0.005 |
| | Medium-Replay | $\beta$ | 0.05 |
| | Random | $\beta$ | 0.05 |

## D.6. Train OMSD Models

In this subsection, we provide the hyperparameters for training OMSD models.

### D.6.1. MPE TASKS

Here are the hyperparameters for all three tasks in `MPE` environments as shown in Table 10.

*Table 10.* OMSD Training Hyperparameters in MPE

| Environment | Task | Hyper Parameter Name | Value |
|---|---|---|---|
| Global | | Training Steps | 100000 |
| | | Annealing Epochs | 10 |
| Cooperative Navigation | Expert | $\beta$ | 0.001 |
| | Medium | $\beta$ | 0.005 |
| | Random | $\beta$ | 0.05 |
| Predator Prey | Expert | $\beta$ | 0.005 |
| | Medium | $\beta$ | 0.05 |
| | Random | $\beta$ | 0.5 |
| World | Expert | $\beta$ | 0.01 |
| | Medium | $\beta$ | 0.05 |
| | Random | $\beta$ | 0.5 |

### D.6.2. MaMuJoCo Tasks

Here are the hyperparameters for MaMuJoCo. The dataset 2-HalfCheetah 200 comes from OMAR (Pan et al., 2022), and the dataset 2-HalfCheetah 210 comes from MADiff (Zhu et al., 2024) as shown in Table 11.

*Table 11.* OMSD Training Hyperparameters in MaMuJoCo

| Environment | Task | Hyper Parameters Name | Value |
|---|---|---|---|
| Global | | Training Steps | 100000 |
| | | Annealing Epochs | 10 |
| 2-HalfCheetah 200 | Expert | $\beta$ | 0.001 |
| | Medium | $\beta$ | 0.005 |
| | Medium-Replay | $\beta$ | 0.05 |
| | Random | $\beta$ | 0.05 |

### D.7. More Ablation Study Results

#### D.7.1. Score Decomposition Methods

Here we present more ablation study results of all three MPE tasks in Fig. 7, i.e., Cooperative Navigation, Predator Prey, and World. Over multiple quality datasets across various tasks, our methods demonstrates advantages over pre-trained Critic IQL and other policy decomposition methods.

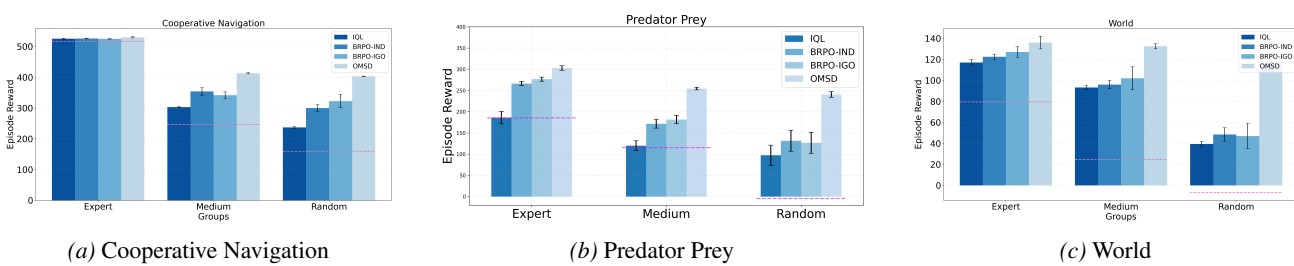

*(a)* Cooperative Navigation     *(b)* Predator Prey     *(c)* World

*Figure 7.* Comparison of Pretrained IQL, BRPO-IND, BRPO-CTDE, and OMSD on Cooperative Navigation, Predator Prey, and World Tasks.

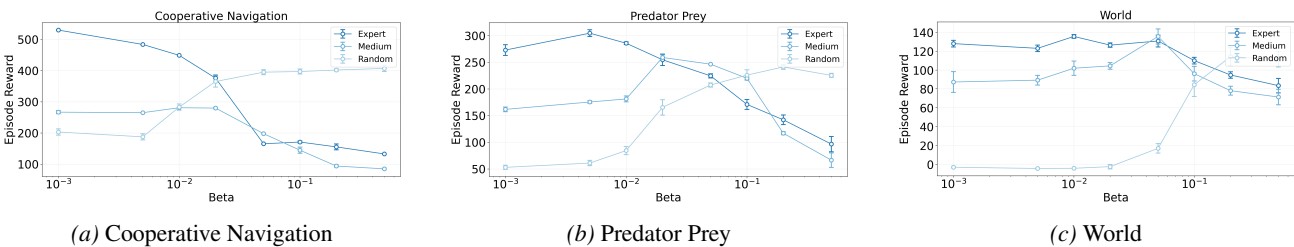

*(a)* Cooperative Navigation        *(b)* Predator Prey        *(c)* World

*Figure 8.* Comparison of regularization term $\beta$ of OMSD on Cooperative Navigation, Predator Prey, and World Tasks.

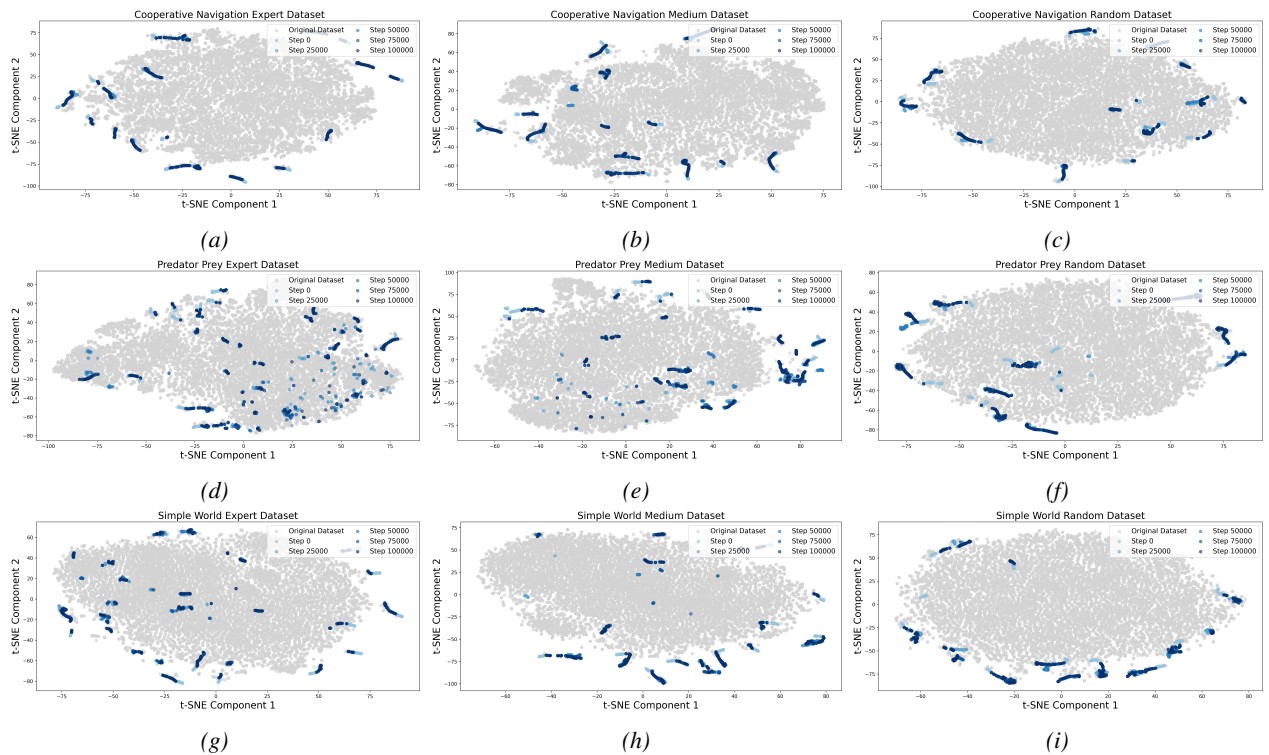

*Figure 9.* Full training trajectories of OMSD on `MPE` tasks.

### D.7.2. HYPERPARAMS

For the temperature coefficient, we sweep over $\beta \in \{0.01, 0.02, 0.05, 0.1, 0.3, 0.5\}$ and observe large variances in appropriate values across different tasks (Fig. 8). We speculate this might be due to $\beta$ being closely intertwined with the behavior distribution and the variance of the Q-value. These factors might exhibit entirely different characteristics across diverse tasks.

### D.7.3. VISUALIZATION OF FINAL POLICY

In Fig. 9, we illustrate the full learning trajectories of OMSD algorithms on `MPE` datasets. The gray data points represent the t-SNE (Van der Maaten & Hinton, 2008) distribution of the state-joint action pairs from the original dataset, while the data points transitioning from light blue to dark blue indicate the t-SNE distribution of episode trajectories collected under policies at different training steps, using 10 random seeds. It can be observed that during the policy update process, the distribution remains mostly within the range of the original dataset, effectively avoiding the OOD problem. This demonstrates that our sequential score decomposition method can encourage that the learning distribution remains in-sample under multimodal offline MARL datasets. Furthermore, as the policy updates, the policy gradually learns and converges to high-reward regions, concentrating within a limited range. This indicates that the joint action critic can effectively provide signals for high-reward regions, guiding policy improvement.

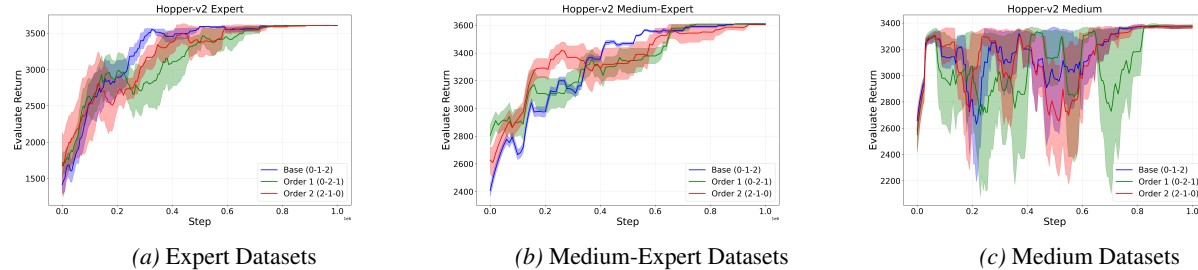

*(a)* Expert Datasets  *(b)* Medium-Expert Datasets  *(c)* Medium Datasets

*Figure 10.* Ablation experiments on three different random update orders of agents in Hopper-v2.

*Table 12.* Density estimator ablation on MPE Predator-Prey. Results are reported as mean ± standard deviation over 5 seeds.

| Dataset | IQL | OMSD-GMM-4 | OMSD-GMM-8 | OMSD-MAF | OMSD-Diffusion |
|---------|-----|------------|------------|----------|----------------|
| Expert  | $100.2 \pm 17.7$ | $68.1 \pm 15.9$ | $60.9 \pm 5.4$ | $49.4 \pm 13.4$ | **$161.4 \pm 9.4$** |
| Medium  | $65.4 \pm 14.1$ | $54.9 \pm 18.6$ | $55.3 \pm 17.7$ | $53.2 \pm 15.5$ | **$137.1 \pm 14.1$** |
| Random  | $53.6 \pm 21.8$ | $88.6 \pm 24.4$ | $86.4 \pm 19.5$ | $114.8 \pm 14.3$ | **$133.9 \pm 16.5$** |

### D.7.4. SEQUENTIAL UPDATE ORDERS

To demonstrate our method's insensitivity to update order, we conducted randomized ordering experiments on the OMIGA Hopper-v2 datasets. Specifically, the task involved three agents. The standard OMSD training process used the default agent ID order as the pre-trained diffusion model and policy update order to determine prefix actions (0-1-2). In addition, we randomly assigned update orders of 0-2-1 and 2-1-0 as control groups to avoid accidental agent relationship modeling under specific update orders. Experimental results show that, with the same pre-training parameters and OMSD training parameter settings, changing only the update order does not significantly impact performance, strongly demonstrating the robustness of our method for capturing complex multimodal behavior distributions. Furthermore, thanks to our structural design, our algorithm only needs to consider the behavior of preceding agents during training, relying solely on its own local observations during execution without needing action information from others. Compared to sequential action modeling methods such as MAT (Wen et al., 2022), this method offers greater flexibility and is insensitive to specific agent dependencies.

### D.7.5. DENSITY ESTIMATOR ABLATION

To examine whether the performance gain of OMSD mainly comes from the sequential conditional decomposition or from the specific choice of diffusion models, we compare diffusion score estimators with simpler conditional density estimators under the same OMSD training pipeline. Specifically, we replace the conditional diffusion model with Gaussian Mixture Models (GMMs, Bishop & Nasrabadi (2006)) and a Masked Autoregressive Flow (MAF, Papamakarios et al. (2017)), while keeping the sequential conditioning structure, pretrained IQL critic, policy architecture, and evaluation protocol unchanged. We evaluate these variants on the MPE Predator-Prey task across expert, medium, and random datasets.

For the GMM variants, we fit agent-wise conditional GMMs with 4 and 8 mixture components, where each model uses the same state and prefix-action conditioning variables as OMSD. For the normalizing-flow variant, we use a standard conditional MAF implementation to estimate the same conditional behavior distribution. The hidden dimension is set to 512, matching the representation scale used by our diffusion score models. The learning rate, batch size, and number of training steps are kept the same as those used for diffusion score-model pretraining. All variants share the same pretrained IQL critic, deterministic policy architecture, policy optimization settings, random seeds, and evaluation protocol as OMSD-Diffusion. Only the conditional behavior score estimator is replaced.

In Table. 12, the results show that the sequential conditional regularization framework can already improve over IQL on the random dataset even with simpler density estimators, suggesting that coordination-aware behavior regularization is useful beyond a particular model class. However, GMMs and MAF perform poorly on expert and medium datasets, where coordinated behaviors are more structured and multimodal. This indicates that inaccurate conditional density estimation can introduce biased regularization directions and hurt policy improvement. We hypothesize that the weak MAF performance is partly due to the difficulty of mapping a connected base distribution to disconnected or weakly connected coordinated modes through continuous bijections, which can assign artificial probability mass to low-density regions. In contrast, diffusion-based score models capture multimodal behavior through denoising score matching and provide consistently

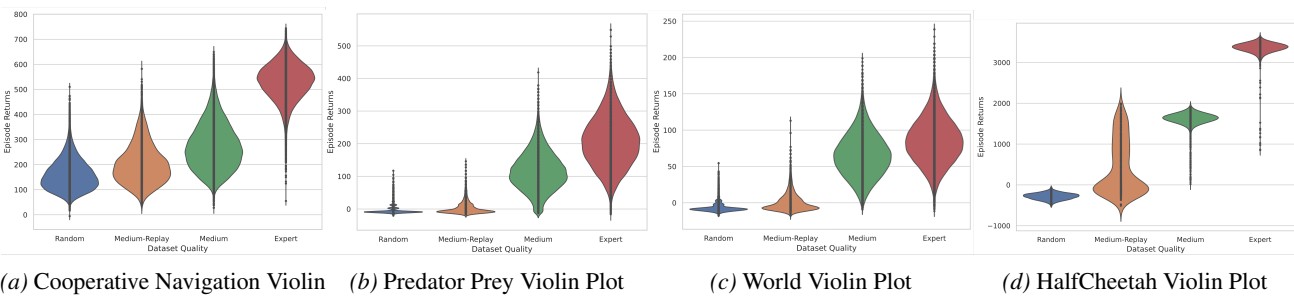

*(a)* Cooperative Navigation Violin Plot  *(b)* Predator Prey Violin Plot  *(c)* World Violin Plot  *(d)* HalfCheetah Violin Plot

*Figure 11.* Violin plots of MPE and `MaMuJoCo` offline datasets.

stronger performance across all data qualities. These results support using diffusion models as the default score estimator in OMSD for complex multimodal offline MARL datasets.

## E. Data Quality Visualization of Offline Datasets

In this section, we provide more details about the offline datasets `MPE`, 2-agent `HalfCheetah` we used in this paper. The data distribution with violin plots are shown in Fig. 11. These plots are provided by OG-MARL[13] (Formanek et al., 2023).

## F. Why do Offline Independent Learning and Naive CTDE Frameworks Fail?

To further elucidate the impact of multimodal behavioral policies on offline MARL, we selected the standard policy-based offline RL method, BRPO (Wu et al., 2019), and extended it to the MARL setting to analyze the failure modes. We focused on two mainstream paradigms: independent learning and CTDE learning.

### F.1. Policy-based Offline MARL with Independent Learning.

We begin our analysis with independent BRPO (BRPO-IND), a fundamental case under the independent learning paradigm. Generally, independent learning methods decompose MARL problems into multiple autonomous single-agent RL processes by treating other agents as part of dynamic environments. This is a robust approach widely adopted in both online and offline MARL algorithms that has demonstrated stable performance across many tasks, which assumes that each policy is independently factorizable. Specifically, in BRPO-IND, each agent independently learns the critic and models individual behavior policy $\mu_i(a_i|s)$ from individual datasets. With Lemma 2.1, we propose the following proposition.

**Proposition F.1.** *Consider a fully cooperative game with n agents. Under the independent learning framework, the optimal individual policy of each agent is:*

$$\pi_i^*(a_i \mid s) = \frac{1}{Z(s)}\mu_i(a_i \mid s)\exp\left(\beta_i Q^i(s, a_i)\right),$$

*where $\mu_i$ and $Q^i$ are individual behavior policy and Q-value function of agent $i$, respectively. With Lemma 2.1, the learning objective of BRPO-IND is:*

$$\mathcal{L}_{Ind} = \max \sum_{i=1}^{n} \mathbb{E}_{s\sim\mathcal{D}_\mu, a_i\sim\pi_{\theta_i}} Q^i(s, a_i) - \underbrace{\frac{1}{\beta}D_{KL}\left[\pi_{\theta_i}\|\mu_i\right]}_{Ind\ Behavior\ Reg}.$$

Here, the KL penalty prevents the learned individual policy from diverging significantly from the individual behavior policy. By taking the gradient of equation $\mathcal{L}_{Ind}$ with respect to each agent's policy parameters, we obtain:

$$\nabla_{\theta_i}\mathcal{L}_{Ind} = \mathbb{E}_{s\sim\mathcal{D}^\mu}\left[\nabla_{a_i}Q^i(s, a_i)\big|_{a_i=\pi_{\theta_i}} + \frac{1}{\beta}\underbrace{\nabla_{a_i}\log\mu_i(a_i \mid s)\big|_{a_i=\pi_{\theta_i}(s)}}_{=-\epsilon_i^*(a_t|s,t)/\sigma_t|_{t\to 0}}\right]\nabla_{\theta_i}\pi_{\theta_i}(a_i|s), \tag{7}$$

---

[13]https://github.com/instadeepai/og-marl

where $\epsilon_i^* (a_t \mid s, t)$ represents the score function of individual behavior policy $\nabla_{a_i} \mu_i(a_i|s)$ (Song et al., 2021).

## F.2. Policy-based Offline MARL with CTDE Learning.

In the CTDE framework, the centralized training process typically leverages the actions of other agents, global states, and the policies of other agents to learn the optimal joint policy. It can stabilize nonstationary learning process by capture interactive relationships between agents and global information. The executable individual policies are usually distilled through value decomposition or policy decomposition. In policy-based methods, such as FOP (Zhang et al., 2021b) and AlberDICE (Matsunaga et al., 2023), the decomposable assumption IGO (Individual-Global-Optimal) $\boldsymbol{\pi}_\Psi^* := \pi_{\psi^i}^{i*} \prod_{j=-i} \pi_{\psi^j}^{j*}$ is typically used to extract individual policies from the joint optimal policy. Based on IGO principle and Lemma 2.1, we propose the BRPO-CTDE as follows.

**Proposition F.2.** *Consider a fully cooperative game with n agents. In centralized learning process, the optimal joint policy is derived as*

$$\pi^*(\boldsymbol{a} \mid s) = \frac{1}{Z(s)} \mu(\boldsymbol{a} \mid s) \exp\left(\beta Q^{tot}(s, \boldsymbol{a})\right),$$

*where $\boldsymbol{a}$ represents the joint actions and $Q^{tot}$ represents the global state-action value function. With Lemma 2.1 and the factorization principle, the learning objective for each agent becomes*

$$\mathcal{L}_{CTDE}^i = \min_{\theta_i} \mathbb{E}_{s \sim \mathcal{D}_\mu} D_{\mathrm{KL}}[\pi_{\theta^i}(\cdot \mid s)\pi_{\theta^{-i}}(\cdot \mid s)||\pi^*(\cdot \mid s)]$$

$$= \max_{\theta_i} \mathbb{E}_{s \sim \mathcal{D}^\mu, \boldsymbol{a} \sim \pi_\theta(\cdot|s)} Q^{tot}(s, \boldsymbol{a}) - \frac{1}{\beta} \underbrace{D_{\mathrm{KL}}\left[\pi_{\theta^i}(\cdot \mid s)\pi_{\theta^{-i}}(\cdot \mid s)||\mu(\boldsymbol{a} \mid s)\right]}_{\textit{Joint Behavior Reg}}.$$

Compared to BRPO-IND, BRPO-CTDE minimizes the KL divergence between the learned joint policy $\Pi_i^n \pi_i(a_i|s)$ and the joint behavior policy distribution $\mu(\boldsymbol{a}|s)$ on each agent's policy update. Then we can derive the gradient of equation $\mathcal{L}_{CTDE}^i$ with respect to each agent's policy parameters as:

$$\nabla_{\theta_i} \mathcal{L}_{CTDE}^i = \mathbb{E}_{s \sim \mathcal{D}^\mu, a^{-i} \sim \pi_{\theta_{-i}}} \left[ \nabla_{a_i} Q^{tot}(s, \boldsymbol{a})\big|_{a=\pi_\theta(\cdot|\boldsymbol{s})} + \frac{1}{\beta} \nabla_{a_i} \log \mu(\boldsymbol{a} \mid s)\big|_{a_i=\pi_{\theta_i}(s)} \right] \nabla_{\theta_i} \pi_{\theta_i}(a_i|s). \tag{8}$$

Equations (7) and (8) reveal that the gradients in offline policy-based MARL consist of Q-value gradients and behavior policy regularization terms. However, this structure poses significant challenges for joint policy updates.

First, an obvious problem arises in the coordination of Q-value gradients. In offline MARL, the absence of online data collection severely limits the ability to adjust policies by exploring new experiences. This issue further exacerbates the misalignment coordination of individual Q-value gradients in MARL and may lead to suboptimal gradient directions (Kuba et al., 2022; Pan et al., 2022).

Admittedly, the CTDE frameworks can slightly alleviate the Q-value gradients coordination problem by directly providing local gradients of the joint Q-function to each agent. However, the individual regularization terms are also challenging due to the multimodal property of the joint behavior policy $\mu(\boldsymbol{a}|s)$. With IGO assumption, the individual behavior regularization term in CTDE becomes a biased score function as

$$\nabla_{a_i} \log \mu(a \mid s) = \nabla_{a_i} \pi(a|s) \nabla_a \log \mu(\boldsymbol{a} \mid s)$$
$$\neq \nabla_{a_i} \log \mu(a^i \mid s),$$

where $\nabla_\pi \log \mu(a|s)$ represents the score function of the joint behavior policy captured by high-capacity generative models, and $\nabla_{a_i} \pi$ is the partial gradient of the joint policy with respect to agent $i$. The primary difficulty lies in accurately calculating $\nabla_{a_i} \pi$ from the multimodal joint behavior policy, as the offline joint policy may not be easily factorizable into individual agent policies.

These challenges faced by BRPO-IND and BRPO-CTDE are fundamentally rooted in the multimodality problem described in Section 3.1 and can be generalized to other policy-based offline RL algorithms. Multimodal joint behavior policies cause complex dependencies among agents, while the infactorization property prevents accurate factorization of these joint policies. Directly applying assumptions in online MARL, such as the factorization assumption, will induce biased policy regularization on individual policy update, ultimately causing the joint policy distribution to deviate from the support set of the dataset.

# G. Theorem Details

## G.1. Proof of Proposition 3.1

We consider a fully-cooperative n-player game with a single state and action space $A = [0, 1]^n$. Let $\pi^*$ be the optimal joint policy with two optimal modes: $a_1 = (1, \ldots, 1)$ and $a_2 = (0, \ldots, 0)$. Let $\hat{\pi}$ be a factorized approximation of $\pi^*$ such that $\hat{\pi}(a) = \prod_{i=1}^n \hat{\pi}_i(a_i)$, where each $\hat{\pi}_i$ is learned independently.

Given that $\pi^*$ has two optimal modes $(1, \ldots, 1)$ and $(0, \ldots, 0)$, and each $\hat{\pi}_i$ is learned independently, the best approximation for each individual policy is to assign equal probability to 0 and 1. Thus, each $\hat{\pi}_i$ converges to Uniform$(\{0, 1\})$, with $\hat{\pi}_i(0) = \hat{\pi}_i(1) = 0.5$ for all $i$.

Since each $\hat{\pi}_i$ is Uniform$(\{0, 1\})$, the joint policy $\hat{\pi}$ will have a mode for each possible combination of 0s and 1s across the $n$ players. There are $2^n$ such combinations. The probability of each mode is $\hat{\pi}(a) = \prod_{i=1}^n \hat{\pi}_i(a_i) = (0.5)^n = 2^{-n}$. Therefore, the reconstruction of joint policy $\hat{\pi}$ exhibits $2^n$ modes, each with probability $2^{-n}$.

To prove that the total variation distance between $\pi^*$ and $\hat{\pi}$ is $\delta_{TV}(\pi^*, \hat{\pi}) = 1 - 2^{1-n}$, we start with the definition of total variation distance:

$$\delta_{TV}(\pi^*, \hat{\pi}) = \frac{1}{2} \sum_a |\pi^*(a) - \hat{\pi}(a)|$$

For $\pi^*$, we have $\pi^*(a_1) = \pi^*((1, \ldots, 1)) = 0.5$, $\pi^*(a_2) = \pi^*((0, \ldots, 0)) = 0.5$, and $\pi^*(a) = 0$ for all other $a$. For $\hat{\pi}$, we have $\hat{\pi}(a) = 2^{-n}$ for all $2^n$ modes.

Calculating the sum of absolute differences:

$$|\pi^*(a_1) - \hat{\pi}(a_1)| + |\pi^*(a_2) - \hat{\pi}(a_2)| = |0.5 - 2^{-n}| + |0.5 - 2^{-n}| = 1 - 2^{1-n}$$

For the remaining $2^n - 2$ modes of $\hat{\pi}$:

$$\sum |0 - 2^{-n}| = (2^n - 2) \cdot 2^{-n} = 1 - 2^{1-n}$$

Therefore,

$$\delta_{TV}(\pi^*, \hat{\pi}) = \frac{1}{2} \cdot (1 - 2^{1-n} + 1 - 2^{1-n}) = 1 - 2^{1-n}$$

As $n \to \infty$, we have:

$$\lim_{n\to\infty} \delta_{TV}(\pi^*, \hat{\pi}) = \lim_{n\to\infty} (1 - 2^{1-n}) = 1 - \lim_{n\to\infty} 2^{1-n} = 1 - 0 = 1$$

This limit indicates a severe distribution shift between the true optimal policy $\pi^*$ and its factorized approximation $\hat{\pi}$ as the number of players increases.

## G.2. Structural Objective Mismatch under Marginal Regularization

We formalize the intuition in Sec. 3.1 as a structural objective mismatch. Marginal behavior regularization constrains the learned joint policy toward the product of individual marginals, $\prod_{i=1}^n \mu_i^{\text{ind}}(a_i|s)$, rather than the true joint behavior distribution $\mu(\boldsymbol{a}|s)$. For a fixed policy $\pi$, this mismatch can be written as

$$\Delta_{\text{CMS}}(\pi) = D_{\text{KL}}\left(\pi \middle\| \prod_{i=1}^n \mu_i^{\text{ind}}\right) - D_{\text{KL}}(\pi\|\mu) = \sum_{i=1}^n \mathbb{E}_{\boldsymbol{a}\sim\pi}\left[\log\frac{\mu_i(a_i|s, \boldsymbol{a}_{<i})}{\mu_i^{\text{ind}}(a_i|s)}\right], \tag{9}$$

where $\mu_i(a_i|s, \boldsymbol{a}_{<i})$ denotes the conditional behavior distribution induced by the chain rule, and $\mu_i^{\text{ind}}(a_i|s)$ denotes the marginal behavior model used by independent regularization. This quantity measures the gap between the intended joint behavior regularizer and its product-of-marginals surrogate. In the balanced two-mode construction of Proposition 3.1, this gap equals $(n-1)\log 2$ when evaluated on the coordinated two-mode target distribution, illustrating that the mismatch can

grow with the number of agents. Sequential decomposition removes this structural factorization error at the behavior-policy level. By the chain rule,

$$\mu(\boldsymbol{a}|s) = \prod_{i=1}^{n} \mu_i(a_i|s, \boldsymbol{a}_{<i}). \tag{10}$$

For the factorized execution policy $\pi(\boldsymbol{a}|s) = \prod_i \pi_i(a_i|s)$, the corresponding joint KL can be decomposed as

$$D_{\mathrm{KL}}(\pi\|\mu) = \sum_{i=1}^{n} \mathbb{E}_{\boldsymbol{a}_{<i}\sim\pi} \left[ D_{\mathrm{KL}}(\pi_i(\cdot|s) \,\|\, \mu_i(\cdot|s, \boldsymbol{a}_{<i})) \right]. \tag{11}$$

Thus, the conditional reference used by OMSD targets the true chain-rule factors of the joint behavior policy rather than independent marginals. This identity shows that sequential decomposition introduces no structural factorization error at the behavior-policy level. Practical errors can still arise from finite data and learned conditional score-model approximation. In the idealized case where each conditional behavior estimator accurately resolves the modes, prefix conditioning restricts later agents toward modes compatible with earlier sampled actions. This reduces cross-mode action combinations that arise under independent marginal regularization. In practice, this property depends on the quality of the learned conditional score models.

### G.3. Proof of Proposition F.1

First, we derive the optimization objectives with independent learning framework. By decomposing the KL term in (F.1), we have

$$\mathcal{L}_{Ind} = \sum_{i=1}^{n} \left( \mathbb{E}_{s\sim\mathcal{D}_\mu, a_i\sim\pi_{\theta_i}} Q^i(s, a_i) + \frac{1}{\beta} \mathbb{E}_{s\sim\mathcal{D}^\mu, a_i\sim\pi_{\theta_i}} \log\mu_i(a_i|s) + \frac{1}{\beta} \mathbb{E}_{s\sim\mathcal{D}^\mu} \mathcal{H}(\pi_i(a_i|s)) \right)$$

where $\mathcal{H}(\pi_i(a_i|s))$ is the entropy of the agent $i$'s policy. As BRPO-IND learns behavior policy independently, we can directly get the term $\log\mu_i(a_i|s)$ implicitly from the pretrained diffusion models of each agent.

Consider that each agent's policy is trained independently without dependency, we can derive the gradient of agent $i$ as

$$\nabla_{\theta_i}\mathcal{L}_{Ind} = \nabla_{\theta_i} \sum_{i=1}^{n} \left( \mathbb{E}_{s\sim\mathcal{D}_\mu, a_i\sim\pi_{\theta_i}} Q^i(s, a_i) + \frac{1}{\beta} \mathbb{E}_{s\sim\mathcal{D}^\mu, a_i\sim\pi_{\theta_i}} \log\mu_i(a_i|s) + \frac{1}{\beta} \mathbb{E}_{s\sim\mathcal{D}^\mu} \mathcal{H}(\pi_i(a_i|s)) \right)$$

$$= \mathbb{E}_{s\sim\mathcal{D}_\mu, a_i\sim\pi_{\theta_i}} \left[ \nabla_{\theta_i} Q^i(s, a_i) + \frac{1}{\beta} \nabla_{\theta_i} \log\mu_i(a_i|s) \right]$$

$$= \mathbb{E}_{s\sim\mathcal{D}^\mu, a_i\sim\pi_{\theta_i}} \left[ \nabla_{\theta_i}\pi_i * \nabla_{a_i} Q^i(s, a_i) + \frac{1}{\beta} \nabla_{\theta_i}\pi_i * \nabla_{a_i} \log\mu_i(a_i|s) \right]$$

$$= \mathbb{E}_{s\sim\mathcal{D}^\mu, a_i\sim\pi_{\theta_i}} \left[ \nabla_{a_i} Q^i(s, a_i) + \frac{1}{\beta} \nabla_{a_i} \log\mu_i(a_i|s) \right] \nabla_{\theta_i}\pi_i.$$

Notice that the term $\nabla_{a_i} \log\mu_i(a_i|s)$ serves as the score function of the independent behavior policy, we can further construct a surrogate loss $\mathcal{L}_{Ind}^{surr}$ and derive a practical gradient for BRPO-IND. Our proof is mainly inspired by the following Lemma G.1.

**Lemma G.1** (Proposition 1 in (Chen et al., 2024)). *Given that $\pi$ is sufficiently expressive, for any time $t$, any state $s$, we have*

$$\arg\min_\pi D_{KL}[\pi_t(\cdot|s)\|\mu_t(\cdot|s)] = \arg\min_\pi D_{KL}[\pi(\cdot|s)\|\mu(\cdot|s)],$$

*where both $\mu_t$ and $\pi_t$ follow the same predefined diffusion process in $q_{t_0}(x_t|x_0) = \mathcal{N}(x_t|\alpha_t x_0, \sigma_t^2 I)$, which implies $x_t = \alpha_t x_0 + \sigma_t \varepsilon$.*

The surrogate loss is

$$L_{Ind}^{\mathrm{surr}}(\theta_i) = \mathbb{E}_{s, a_i\sim\pi_{\theta_i}} Q(s, a_i) - \frac{1}{\beta} \mathbb{E}_{t, s} \omega(t) \frac{\sigma_t}{\alpha_t} D_{\mathrm{KL}}[\pi_{\theta_i, t}(\cdot|s)\|\mu_{i, t}(\cdot|s)]. \tag{12}$$

Then we can propose the practical gradient as follows.

**Proposition G.2** (Practical Gradient of BRPO-IND). *Given that $\pi_{\theta_i}$ is deterministic policy and $\epsilon_i^*$ is the optimal diffusion model of independent behavior policy $\mu_i$, the gradient of the surrogate loss (12) w.r.t agent $i$ is*

$$\nabla_{\theta_i} L_{surr}^{\pi}(\theta) = \left[ \mathbb{E}_s \nabla_a Q_\phi(s,a)|_{a=\pi_\theta(s)} - \frac{1}{\beta} \mathbb{E}_{t,s} \omega(t)(\epsilon_i^*(a_{t,i}|s,t) - \epsilon_i)|_{a_{i,t}=\alpha_t \pi_{\theta_i}(s) + \sigma_t \epsilon_i} \right] \nabla_{\theta_i} \pi_{\theta_i}(s).$$

*Proof.* The fundamental framework of the proof follows the proof process of SRPO (Chen et al., 2024), extending it to the multi-agent scenario. Based on the forward diffusion process in section 2.2, we can represent the noisy distribution of actor policy at step $t$ as

$$\pi_{\theta_i,t}(a_{t,i}|s) = \int \mathcal{N}(a_{i,t}|\alpha_t a_i, \sigma_t^2 I) \pi_{\theta_i}(a_i|s) da_i \tag{13}$$

$$= \int \mathcal{N}(a_{t,i}|\alpha_t a_i, \sigma_t^2 I)\delta(a_i - \pi_{\theta_i}(s)) da_i \tag{14}$$

$$= \mathcal{N}(a_{t,i}|\alpha_t \pi_{\theta_i}(s), \sigma_t^2 I) \tag{15}$$

Note that $\pi_{\theta,t}(\cdot|s)$ is a Gaussian policy with expected value $\alpha_t \pi_\theta(s)$ and variance $\sigma_t^2 I$, we can simplify the surrogate training objective as

$$L_{Ind}^{surr}(\theta_i) = \mathbb{E}_{s,a_i \sim \pi_{\theta_i}(\cdot|s)} Q(s,a_i) - \frac{1}{\beta} \mathbb{E}_{t,s} \omega(t) \frac{\sigma_t}{\alpha_t} D_{\mathrm{KL}}[\pi_{\theta_i,t}(\cdot|s) \| \mu_{i,t}(\cdot|s)]$$

$$= \mathbb{E}_s Q(s,a_i)|_{a_i=\pi_{\theta_i}(s)} + \frac{1}{\beta} \mathbb{E}_{t,s} \omega(t) \frac{\sigma_t}{\alpha_t} \mathbb{E}_{a_{i,t} \sim \mathcal{N}(\cdot|\alpha_t \pi_{\theta_i}(s), \sigma_t^2 I)} [\log \mu_t(a_{i,t}|s) - \log \pi_{t,\theta_i}(a_{i,t}|s)]$$

Then we can derive the gradient of this objective as follows

$$\nabla_{\theta_i} \mathcal{L}_{Ind}^{surr}(\theta_i) = \nabla_{\theta_i} \mathbb{E}_{\boldsymbol{s} \sim \mathcal{D}^\mu} Q_\phi(\boldsymbol{s}, a_i)|_{a_i \sim \pi_\theta^i(\boldsymbol{s})} + \frac{1}{\beta} \mathbb{E}_{t,s} \frac{\sigma_t}{\alpha_t} \omega(t) \nabla_{\theta_i} \mathbb{E}_{\epsilon_i} \left[ \log \mu_t^i(a_t^i|s) - \log \pi_t^i(a_t^i|s) \right]$$

$$\text{(reparameterization of } \pi_i = \alpha_t \pi_{\theta_i}(s) + \sigma_t \epsilon_i)$$

$$= \nabla_{\theta_i} \mathbb{E}_{\boldsymbol{s} \sim \mathcal{D}^\mu} Q_\phi(\boldsymbol{s}, a_i)|_{a_i \sim \pi_\theta^i(\boldsymbol{s})} + \frac{1}{\beta} \mathbb{E}_{t,s,\epsilon_i} \frac{\sigma_t}{\alpha_t} \omega(t) \left[ \nabla_{\theta_i} \log \mu_t^i(a_t^i|s) - \nabla_{\theta_i} \log \pi_t^i(a_t^i|s) \right] \quad \text{(chain rule)}$$

$$= \nabla_{\theta_i} \mathbb{E}_{\boldsymbol{s} \sim \mathcal{D}^\mu} Q_\phi(\boldsymbol{s}, a_i)|_{a_i \sim \pi_\theta^i(\boldsymbol{s})} + \frac{1}{\beta} \mathbb{E}_{t,s,\epsilon_i} \frac{\sigma_t}{\alpha_t} \omega(t) \big[ \nabla_{a_i^t} \log \mu_t^i(a_t^i|s) \nabla_{\theta_i} a_i^t|_{a_i^t = \alpha_t \pi_{\theta_i}(s) + \sigma_t \epsilon_i}$$

$$- \nabla_{a_i^t} \log \pi_t^i(a_t^i|s) \nabla_{\theta_i} a_i^t|_{a_i^t = \alpha_t \pi_{\theta_i}(s) + \sigma_t \epsilon_i} \big]$$

$$= \mathbb{E}_{\boldsymbol{s} \sim \mathcal{D}^\mu} \nabla_{a_i} Q_\phi(\boldsymbol{s}, \boldsymbol{a}_i, \boldsymbol{a}_{-i})|_{\boldsymbol{a}_i \sim \pi_\theta^i(\boldsymbol{s}), \boldsymbol{a}_{-i} \sim \pi_\theta^{-i}(\boldsymbol{s})} \nabla_{\theta_i} \pi_i$$

$$+ \frac{1}{\beta} \mathbb{E}_{t,s,\epsilon_i} \frac{\sigma_t}{\alpha_t} \omega(t) \left[ -\frac{\epsilon_i(a_i|s,t)}{\sigma_t} \alpha_t \nabla_{\theta_i} \pi_{\theta_i}(s) + \frac{\epsilon}{\sigma_t} \alpha_t \nabla_{\theta_i} \pi_{\theta_i}(s) \right]$$

$$= \left[ \underbrace{\mathbb{E}_{\boldsymbol{s}} \nabla_{a_i} Q_\phi(\boldsymbol{s}, \boldsymbol{a}_i, \boldsymbol{a}_{-i})|_{\boldsymbol{a}_i \sim \pi_\theta^i(\boldsymbol{s}), \boldsymbol{a}_{-i} \sim \pi_\theta^{-i}(\boldsymbol{s})}}_{\text{Q gradient}} \right.$$

$$\left. - \frac{1}{\beta} \mathbb{E}_{t,s,\epsilon_i} \omega(t) \left( \underbrace{\epsilon_i(a_i^t|s,t)}_{\text{score } \mu_i^t} - \underbrace{\epsilon}_{\text{score } \pi_i^t} \right)|_{a_i^t = \alpha_t \pi_{\theta_i}(s) + \sigma_t \epsilon_i} \right] \nabla_{\theta_i} \pi_i(s)$$

$$\tag{16}$$

$\square$

## G.4. Proof of Proposition F.2

First, we derive the optimization objectives with centralized learning framework. By decomposing the KL term, we have

$$\mathcal{L}_{CTDE}^i = \mathbb{E}_{s \sim \mathcal{D}^\mu, \boldsymbol{a} \sim \pi_\theta(\cdot|s)} Q^{tot}(s, \boldsymbol{a}) + \frac{1}{\beta} \mathbb{E}_{s \sim \mathcal{D}^\mu, \boldsymbol{a} \sim \pi_\theta(\cdot|s)} \log \mu(\boldsymbol{a}|s) + \frac{1}{\beta} \mathbb{E}_{s \sim \mathcal{D}^\mu} \mathcal{H}(\pi(\boldsymbol{a}|s)),$$

where $\mathcal{H}(\pi(\boldsymbol{a}|s))$ is the entropy of the joint policy. Then we need to distill the decentralized executive policy for each agent. Consider that each agent policy $\pi_{\theta_i}$ is an isotropic Gaussian policy, we can decompose the joint policy by $\pi = \pi_{\theta_i}\pi_{\theta_{-i}}$. The gradient of agent $i$ is as follows

$$\nabla_{\theta_i}\mathcal{L}^i_{CTDE} = \nabla_{\theta_i}\mathbb{E}_{s\sim\mathcal{D}^\mu,\boldsymbol{a}_{-i}\sim\pi_{\theta_{-i}}(\cdot|s)}\left[Q^{tot}(s,\boldsymbol{a}) + \frac{1}{\beta}\log\mu(\boldsymbol{a}|s)\right] \tag{17}$$

$$= \mathbb{E}_{s\sim\mathcal{D}^\mu,\boldsymbol{a}_{-i}\sim\pi_{\theta_{-i}}(\cdot|s)}\left[\nabla_{\theta_i}Q^{tot}(s,\boldsymbol{a}) + \frac{1}{\beta}\nabla_{\theta_i}\log\mu(\boldsymbol{a}|s)\right] \tag{18}$$

$$= \mathbb{E}_{s\sim\mathcal{D}^\mu,\boldsymbol{a}_{-i}\sim\pi_{\theta_{-i}}(\cdot|s)}\left[\nabla_{\theta_i}\pi_i * \nabla_{a_i}Q^{tot}(s,\boldsymbol{a}) + \frac{1}{\beta}\nabla_{\theta_i}\pi_i * \nabla_{a_i}\log\mu(\boldsymbol{a}|s)\right] \tag{19}$$

$$= \mathbb{E}_{s\sim\mathcal{D}^\mu,\boldsymbol{a}_{-i}\sim\pi_{\theta_{-i}}(\cdot|s)}\left[\nabla_{a_i}Q^{tot}(s,\boldsymbol{a}) + \frac{1}{\beta}\nabla_{a_i}\log\mu(\boldsymbol{a}|s)\right]\nabla_{\theta_i}\pi_i. \tag{20}$$

Importantly, different from the cases in BRPO-IND, we cannot distill a score function $\nabla_{a_i}\log\mu(\boldsymbol{a}|s)$ from the pretrained diffusion models of joint behavior policies. To illustrate the influence of inappropriate factorizations, we slightly abuse the factorization assumptions to decompose the joint behavior policy as $\mu(\boldsymbol{a}|s) = \prod_{i=1}^n \mu_i(a_i|s)$ and propose a revised baseline called BRPO-CTDE. This variant shares most of the framework with BRPO-CTDE, but differs in the policy regularization component: instead of using the joint behavior policy, BRPO-CTDE employs individual behavior policies for regularization.

## H. Details about Practical Algorithm

### H.1. OMSD Pipeline

The OMSD methods contain a two-stage training process: 1) pretraining sequential diffusion models and joint action critic on the dataset by making score decomposition, and 2) injecting decomposed scores as the individual policy regularization terms into the critic and derive deterministic policies for execution. The resulting OMSD algorithm is presented in Algorithm 1.

The basic workflow of OMSD follows the idea of SRPO (Chen et al., 2024) by extending the single agent learning process into multi-agent process, where the sequential conditional score decomposition proposed in section 3.2 is plugged in to reduce the miscoordinated policy updates. Specifically, as we take the joint critic and individual score regularization, all the agents share the copies of a pre-trained common joint action Q-networks $Q_{tot}$ and keep individual pre-trained behavior diffusion models to extract the score regularization. This is a common setup in multi-agent reinforcement learning, such as MADDPG. Besides, each agent maintains a deterministic policy as the actor network, which bypasses the heavy iterative denoising process of diffusion models to generate actions and enjoy the fast decision-making speed.

### H.2. Pretraining IQL as Critic

The centralized Q-network are pretrained with implicit Q-learning (Kostrikov et al., 2022), which introduced the expectile regression in pessimistic value estimation:

$$\min L_V(\zeta) = \mathbb{E}_{(s,a)\sim\mathcal{D}_\mu}\left[L_2^\tau(Q_\phi(s,a) - V_\zeta(s))\right],$$
$$\min L_Q(\phi) = \mathbb{E}_{(s,a,s')\sim\mathcal{D}_\mu}\left[||r(s,a) + \gamma V_\zeta(s') - Q_\phi(s,a)||_2^2\right],$$

where $L_2^\tau(u) = |\tau - \mathbf{1}(u < 0)|u^2$ is the expectile operator.

### H.3. Pretraining Diffusion Models

Considering the state and actions are continuous, the behavior models are trained with classifier-free guidance diffusion models (Hansen-Estruch et al., 2023; Chen et al., 2024) by minimizing the following loss:

$$\min_{\psi_i} L_\mu(\psi_i) = \mathbb{E}_{t,\epsilon_i,(s,a)\sim\mathcal{D}_\mu}\left[||\hat{\epsilon}_{\psi_i}(a_t^i|s, a^{i-}, t) - \epsilon||_2^2\right]_{a_t^i = \alpha_t a^i + \sigma_t \epsilon}, \tag{21}$$

where $t \sim \mathcal{U}(0,1), \epsilon \sim \mathcal{N}(0,\mathbf{I})$, and the sequential score function can be estimated with $\hat{\epsilon}_{\psi_i}(a_t^i|s, a^{i-}, t) \approx -\sigma_t\nabla_{a_i}\log\mu(a_i|s, a^{i-})$ (Song et al., 2021).

Following similar numerical computation simplification methods in SRPO (Chen et al., 2024), we also utilize the intermediate distributions of the entire diffusion process $t \in [0, 1]$ to replace the original training objective here. The surrogate objective is

$$\max_{\theta_i} \mathcal{L}_\pi^{surr}(\theta_i) = \mathbb{E}_{\boldsymbol{s} \sim \mathcal{D}^\mu, \boldsymbol{a_i} \sim \pi_i(\cdot|s), \boldsymbol{a_{-i}} \sim \pi_{-i}(\cdot|s)} Q_\phi(\boldsymbol{s}, \boldsymbol{a_i}, \boldsymbol{a_{-i}}) \tag{22}$$
$$- \frac{1}{\beta} \mathbb{E}_{t,s} \omega(t) \frac{\sigma_t}{\alpha_t} D_{\mathrm{KL}} \left[ \pi_{i,t}(\cdot \mid \boldsymbol{s}) \| \mu_{i,t}(\cdot \mid \boldsymbol{s}, a^{i-}) \right] |_{a^{i-} \sim \pi^{i-}},$$

where $\omega(t) = \delta(t - 0.02) \frac{\alpha_{0.02}}{\sigma_{0.02}}$ is the weighting parameters to ensure the gap between $\mathcal{L}^{surr}(\theta_i)$ and $\mathcal{L}(\theta_i)$, $\pi_{i,t}(\cdot \mid \boldsymbol{s}) := \mathbb{E}_{a_i \sim \pi_i(\cdot|s)} \mathcal{N}(a_{i,t}|\alpha_t a_i, \sigma_t^2 \mathbf{I})$, and $\mu_{i,t}(\cdot \mid \boldsymbol{s}, a^{i-}) := \mathbb{E}_{a_i \sim \mu_{i,t}(\cdot|s, a^{i-})} \mathcal{N}(a_{i,t}|\alpha_t a_i, \sigma_t^2 \mathbf{I})$.

Considering the instability of the diffusion model near the initial and terminal times, we truncate the time range as $t \sim \mathcal{U}(0.02, 0.98)$. Therefore, we can derive the practical gradients for optimizing the objective as

$$\nabla_{\theta_i} \mathcal{L}_\pi(\theta_i) = \mathbb{E}_{\boldsymbol{s} \sim \mathcal{D}^\mu, a^{i-} \sim \bar{\pi}^{-i}, a^{i+} \sim \pi^{i+}} [\nabla_{\boldsymbol{a_i}} Q_\phi(\boldsymbol{s}, \boldsymbol{a})|_{a_i = \pi_{\theta_i}, a_{-i} = \pi_{\theta_{-i}}(\boldsymbol{s})} \tag{23}$$
$$+ \frac{1}{\beta} \underbrace{\nabla_{\boldsymbol{a_i}} \boldsymbol{a} \cdot \nabla_{\boldsymbol{a}} \log \mu(\boldsymbol{a} \mid \boldsymbol{s})|_{\boldsymbol{a} = \boldsymbol{\pi_\theta}(\boldsymbol{s})}}_{= -\boldsymbol{\epsilon}^*(\boldsymbol{a_t}|\boldsymbol{s}, t)/\sigma_t|_{t \to 0}} ] \nabla_{\theta_i} \pi_{\theta_i}(\boldsymbol{s}).$$

Compared to the naive score decomposition methods BRPO-CTDE, the main improvement is replacing the biased score regularization with sequential decomposed score. It strongly promotes the policy update directions and coordination among all agents' gradients.

### H.4. Discussions

In OMSD, the sequential conditional distribution is solely utilized during the policy update phase to extract conditional score functions for policy regularization. Specifically, the sequential structure is not embedded in the execution policy. Instead, it is only used to model the joint behavior policy and derive score functions that guide individual policy updates. This design ensures that during execution, each agent's policy remains independently executable based solely on local observations, without requiring sequential action selection or global coordination at runtime.

In continuous control tasks, the policy is typically modeled as a Dilac distribution (or Gaussian distribution). Without loss of generality, we employ the Dilac policy, which provides deterministic prefix actions $a_{<i}$ given the state during the policy update of agent $i$. This approach not only preserves the flexibility of simultaneous decision-making but also enables efficient parallel pre-training of score models for each agent directly from the dataset. By decoupling the sequential modeling of joint behavior policies from the execution phase, OMSD achieves a unique balance between coordinated learning and decentralized execution, making it highly efficient and scalable for real-world multi-agent scenarios.

While Gaussian policies are standard in continuous control, they are suboptimal for sequential score regularization since sampling stochastic prefix actions causes noise propagation and instability. Instead, we adopt Dilac policies—deterministic mappings with likelihood approximation capacity—to ensure that prefix actions remain stable and deterministic during training. This design choice aligns with the score distillation requirement and allows high-throughput parallel updates across agents, improving both training efficiency and scalability.

Crucially, OMSD does not employ the diffusion model as an actor network during execution, which could lead to out-of-distribution (OOD) action problems due to the iterative sampling process (Mao et al., 2024). Instead, we only perturb the sampled actions from policy $a_i^0 = \pi(a_i|s)$ with a random noise $\epsilon_t \sim \mathcal{N}(\mathbf{0}, \boldsymbol{I})$ to construct latent variables $a_i^t$ and use the diffusion model to compute the corresponding score function $\hat{\epsilon}(a_i^t|s, a_{<i}, t)$ as behavior regularization. This approach avoids the computationally expensive ancestral sampling required in denoising steps in traditional diffusion models, significantly accelerating both training and execution.

Figure 3 illustrates the training workflow of OMSD. Joint offline data is reused to train a global Q function $Q^{tot}(s, \boldsymbol{a})$ and agent-wise conditional diffusion models. During policy updates, each agent receives two types of guidance: **top-down guidance** from $Q^{tot}(s, \boldsymbol{a})$, for identifying high-value regions, and **bottom-up score regularization** from the diffusion model, which conditions on prior agents' actions and regularizes against OOD updates.

This two-way information flow enables coordinated learning while encouraging in-distribution updates at each step. Even when earlier agents' policies deviate, the proper conditional score guides corrections, preserving a stable joint behavior

pattern. Moreover, because diffusion models are used only for score estimation, not sampling, OMSD avoids diffusion-based actor workflows that suffer from iterative sampling inefficiency and OOD action generation (Mao et al., 2024). The final policies remain lightweight, independently executable, and deployable in fully decentralized environments.

# I. Computational Resources

For `MaMuJoCo` and `MPE` experiments, we utilized a single NVIDIA Geforce RTX 3090 graphics processing unit (GPU). For the most complex `MaMuJoCo` task, training IQL takes 6-10 hours, training the diffusion model for each agent takes 4-6 hours, and training the OMSD policy update only takes 1-2 hours to converge. For the simpler tasks such as MPE and bandit, each module only takes 1 hour and 10 minutes respectively. Since the sequential diffusion model for each agent can be trained in parallel using the data from the dataset, multiple pretraining models can be initiated in parallel to avoid the training time increasing linearly with the number of agents.

# J. Use of LLMs

We use LLMs for polish writing. Specifically, LLMs assist in refining the grammar, clarity, and overall presentation of the paper, ensuring that the text is clear and professionally written. No experimental results or core content were generated by LLMs.

# K. Ethics Statement

Our work does not involve human subjects, sensitive data, or personally identifiable information. The research is purely theoretical and is not expected to raise any ethical concerns. All experiments are conducted in simulated environments and comply with the relevant ethical guidelines of our institution.

# L. Reproducibility Statement

We provide all the details necessary to reproduce our results. The main paper and supplementary materials contain a comprehensive description of the model architecture, training procedure, and hyperparameters. Our source code is available at https://github.com/qiaodan-cuhk/OMSD.

