# OpenReview forum: "Offline Multi-Agent Reinforcement Learning via Sequential Score Decomposition"
_ICML.cc/2026/Conference — ICML 2026 regular_

### Official Review · Reviewer_jcJo · 2026-03-09

**Soundness:** 3
**Presentation:** 3
**Significance:** 3
**Originality:** 3
**Overall Recommendation:** 4
**Confidence:** 3

**Summary:**

This paper proposes OMSD, a sequential score decomposition framework for offline cooperative MARL. The method models joint actions through sequential conditional decomposition with diffusion-based score estimation, and demonstrates strong empirical performance on several continuous-control benchmarks.

**Compliance With Llm Reviewing Policy:**

Affirmed.

**Final Justification:**

The authors have addressed my concern, and I raised my score.

**Key Questions For Authors:**

1. Can you empirically or theoretically compare OMSD against simpler, more computationally efficient density estimators (such as Gaussian Mixture Models or normalizing flows) to fundamentally justify the algorithmic necessity of high-capacity diffusion models for this specific offline MARL setting?

**Limitations:**

Yes

**Strengths And Weaknesses:**

**Strengths:**
- A notable strength of the paper is its application of a score-based diffusion model to address the Combinatorial Mode Shift (CMS) problem, enabling the policy to capture multimodal cooperative behaviors from offline data.
- Empirical results on the MPE environment indicate clear performance improvements over state-of-the-art baselines.

**Weaknesses:**
- While the authors formalize the Combinatorial Mode Shift (CMS) problem through Proposition 3.1, the result only illustrates the failure of independent factorization in a highly stylized toy setting—an n-player game over $[0,1]^n$ with exactly two extreme corner modes. Moreover, the manuscript does not provide a corresponding rigorous proof showing that the proposed solution, Sequential Score Decomposition (SSD), can effectively prevent exponential mode growth in general continuous environments with complex, overlapping multimodal distributions. However, I fully understand its difficulty.
- The manuscript does not provide a sufficiently strong justification for the methodological complexity introduced by using high-capacity diffusion models solely for behavior regularization. The paper lacks systematic empirical comparisons or theoretical arguments showing that simpler, more computationally efficient density estimators would be inadequate for capturing the necessary dependencies in this offline MARL setting. Without evaluating the proposed approach against such alternatives, the use of a diffusion-based architecture appears unnecessarily complex for the intended purpose.

---

> ### Author Rebuttal · Authors · 2026-03-31
>
> We sincerely thank the reviewer for the detailed review and for recognizing OMSD's clear performance improvements on the MPE benchmarks. Regarding your questions:
>
> **W1: On Proposition 3.1 as a Toy Setting**
>
> We agree that Proposition 3.1 is a stylized and illustrative example. As the reviewer indicated, providing a strict mathematical proof to prevent exponential mode growth in high-dimensional continuous spaces with neural network approximations is indeed an open challenge in the field.
>
> However, our primary goal with Proposition 3.1 is not to provide a global convergence proof, but rather, for mathematical conciseness, to most clearly reveal the pathology of CMS problems in offline multimodal MARL. We are also encouraged that Reviewer mCGe explicitly validated our analysis, noting that Fig. 1 and 2 do the "real explanatory work," and Reviewer 8rze also agreed that our motivation and formalization of the multimodal issue are highly clear.
>
> To bridge the gap between this "minimalist theory" and the "complex, overlapping multimodal distributions" you correctly pointed out, we specifically simulated overlapping high-dimensional multimodal scenarios in our 2-GMM Bandit experiment in Sec 4.1. The empirical results clearly demonstrate the stability and effectiveness of our SSD in handling such overlapping distributions. In the revision, we will temper some of our absolute theoretical claims and further emphasize how the t-SNE visualizations (Fig. 4c) and the Bandit experiment serve as empirical support and natural extrapolations of the theory.
>
> **W2 & Q1: Simpler Estimators**
>
> We appreciate this constructive question. We first want to clarify our core insights: our primary contribution is identifying the CMS problem and proposing the coordinated regularization framework. Diffusion models are one class of high-capacity conditional score estimators that satisfy this framework's requirements. Theoretically, other explicit density models (e.g., GMMs or Normalizing Flows) could also provide $\nabla_a \log p(a|s)$ for behavior regularization.
>
> However, we engineered OMSD using diffusion models because they have emerged as the state-of-the-art choice for modeling complex, multimodal distributions in recent years, owing to their superior expressive power and training stability. As noted by Reviewers qh4B and mCGe, OMSD excels not only on MPE but also achieves breakthrough improvements on highly challenging high-dimensional continuous control tasks like MaMuJoCo (e.g., 6-HalfCheetah, 3-Hopper).
>
> To empirically directly answer your question, we conducted an ablation study using GMM-MARL. We trained GMMs with different numbers of components (4 and 8) as density estimators using the same sequential conditional modeling method as OMSD, and strictly tested them on the MPE Predator Prey task across 5 seeds.
>
> | Dataset Quality | Baseline (IQL) | OMSD-GMM-4 | OMSD-GMM-8 | **OMSD-Diffusion [Ours]** |
> | :--- | :--- | :--- | :--- | :--- |
> | **Expert** | 100.2 ± 17.7 | 68.1 ± 15.9 (-32%) | 60.9 ± 5.4 (-39%) | **161.4 ± 9.4 (+61%)** |
> | **Medium** | 65.4 ± 14.1 | 54.9 ± 18.6 (-16%) | 55.3 ± 17.7 (-15%) | **137.1 ± 14.1 (+110%)** |
> | **Random** | 53.6 ± 21.8 | 88.6 ± 24.4 (+65%) | 86.4 ± 19.5 (+62%) | **133.9 ± 16.5 (+150%)** |
>
> The results in the table demonstrate that:
>
> 1. On the highly challenging **Random** dataset, even the simple GMM significantly outperforms the baseline IQL which lacks multimodal coordination capabilities. This suggests that in low-quality data regimes with highly stochastic distributions, even a coarse-grained behavioral constraint can provide essential guidance to prevent severe OOD actions, highlighting the fundamental necessity of our coordinated regularization framework regardless of the backbone.
>
> 2. However, on the **Expert** and **Medium** datasets containing more complex coordinated trajectories, GMMs suffer from mode collapse and limited expressiveness, introducing biased regularization gradients that severely degrade performance. **Notably, GMM-based regularization performs even below the IQL baseline on Expert/Medium data, confirming that inaccurate behavior modeling can be actively harmful rather than merely unhelpful.** In contrast, OMSD-Diffusion achieves significant and stable improvements across all data qualities.
>
> This strongly justifies that introducing high-capacity diffusion models is by no means "over-engineering," but rather an essential component to accurately capture underlying modal distributions in complex MARL environments. We will add these proof-of-concept experiments and discussions on simpler density estimators to the revised appendix.

---

> > ### Author Rebuttal · Reviewer_jcJo · 2026-04-03
> >
> > Thank you for the detailed rebuttal and the additional experiments. While my primary concern has been partially addressed, a few critical points remain unresolved:
> >
> > My original query specifically suggested evaluating against GMMs or Normalizing Flows (NFs). Demonstrating the failure of a simple GMM with 4-8 components on Expert/Medium datasets is informative; however, it is well established that GMMs are notoriously prone to mode collapse in high-dimensional continuous control. NFs offer significantly higher expressive capacity while remaining computationally cheaper than diffusion models. Without an evaluation against a stronger baseline like NFs, the conclusion that high-capacity diffusion models are uniquely 'essential' remains overstated. Could the authors comment on this or provide a comparison?
> >
> > I remain slightly unsatisfied with the theoretical depth of this work. Given the methodological complexity introduced, the paper lacks the necessary theoretical guarantees (or at least some insight) to fully justify the framework.

---

> > > ### Author Response · Authors · 2026-04-05
> > >
> > > We sincerely thank the reviewer for the follow-up feedback. We would like to address your remaining concerns regarding empirical baselines and theoretical depth below.
> > >
> > > **1. Empirical Justification: Normalizing Flows vs. Diffusion Models**
> > >
> > > Based on your suggestion, we have supplemented an ablation study using Normalizing Flows. This experiment strongly supports our core contribution: addressing severe CMS problems in offline MARL necessitates an appropriate sequential decomposition paired with an expressive policy regularization model. A high-capacity model is an architectural necessity, and diffusion models are well-suited for expressing the score functions in OMSD.
> > >
> > > Specifically, we implemented a Masked Autoregressive Flow (MAF), a representative high-capacity NF, as the sequential density estimator. We evaluated it on the MPE Predator-Prey task under the exact same experimental settings. The results are as follows:
> > >
> > > | Dataset Quality | Baseline (IQL) | OMSD-GMM-4 | OMSD-MAF [New] | OMSD-Diffusion [Ours] |
> > > | :--- | :--- | :--- | :--- | :--- |
> > > | **Expert** | 100.2 ± 17.7 | 68.1 ± 15.9 (-32%) | **49.4 ± 13.4 (-51%)** | **161.4 ± 9.4 (+61%)** |
> > > | **Medium** | 65.4 ± 14.1 | 54.9 ± 18.6 (-16%) | **53.2 ± 15.5 (-19%)** | **137.1 ± 14.1 (+110%)** |
> > > | **Random** | 53.6 ± 21.8 | 88.6 ± 24.4 (+65%) | **114.8 ± 14.3 (+114%)** | **133.9 ± 16.5 (+150%)** |
> > >
> > > In the Random dataset, NFs demonstrate stronger expressiveness than GMMs, yet still perform significantly worse than the Diffusion model. However, on the Expert and Medium datasets, MAF suffers a severe performance drop, even underperforming GMMs.
> > >
> > > We hypothesize this performance gap stems from the highly multi-modal nature of expert coordinated behaviors with disjoint supports. Because normalizing flows (e.g., MAF) are constrained to learn a continuous bijection from a connected base distribution, they inevitably assign artificial probability mass to low-density, OOD regions when attempting to cover disconnected modes, severely degrading policy regularization. In contrast, diffusion models avoid these topological limitations, enabling more precise modeling of complex, multi-modal distributions via denoising score matching. Empirical results also support that diffusion models are the most appropriate design for our framework.
> > >
> > > **2. Addressing the Theoretical Depth Concern**
> > >
> > > We also appreciate your push for stronger theoretical justification. We now formalize our core insight to fully justify the framework.
> > >
> > > The fundamental issue identified in Proposition 3.1 is that independent methods optimize a different and biased objective from the intended behavior-regularized joint policy optimization. As formalized in Proposition 3.1, under a multi-agent setting with $n$ agents and multimodal behavior policies, this introduces an irreducible objective bias $\Delta_{\text{CMS}}$:
> > >
> > > $\Delta_{\text{CMS}} = D_{KL}\left(\pi \Big\| \prod_{i=1}^n \mu_i^{\text{ind}}\right) - D_{KL}(\pi \| \mu) = \sum_{i=1}^n E_{\mathbf{a}\sim\pi}\left[\log \frac{\mu_i(a_i|s, a_{<i})}{\mu_i^{\text{ind}}(a_i|s)}\right]$
> > >
> > > In the exact multimodal setting of Proposition 3.1, evaluating this difference yields $\Delta_{\text{CMS}} = (n-1)\log 2$. This objective bias propagates into policy optimization and scales linearly with the number of agents $n$, explaining why independent methods systematically fail as the multi-agent system grows.
> > >
> > > OMSD resolves this via sequential decomposition. By modeling $\mu(\mathbf{a}|s) = \prod_i \mu_i(a_i|s, a_{<i})$, the KL chain rule gives the exact identity:
> > >
> > > $$D_{KL}(\pi \| \mu) = \sum_{i=1}^n E_{a_{<i}\sim\pi}\left[D_{KL}(\pi_i( \cdot |s, a_{<i}) \| \mu_i( \cdot |s, a_{<i}))\right]$$
> > >
> > > Theoretically, our sequential decomposition ensures there is no structural approximation error in the objective function ($\Delta_{\text{CMS}} = 0$). While practical estimation error in the learned score functions remains, this framework correctly aligns the fundamental optimization target.
> > >
> > > Furthermore, the conditional structure ensures support consistency: once $a_{<i}$ falls within a specific mode, $\mu_i(\cdot|s, a_{<i})$ collapses to that mode's support, preventing cross-mode combinations—preserving $K$ modes rather than generating $K^n$ spurious ones. Crucially, this support-consistency property requires the density estimator to accurately resolve the conditional multimodal structure; otherwise, the conditional distribution may not collapse to a single mode.
> > >
> > > We emphasize that this theoretical guarantee holds at the framework level, independent of the chosen density estimator. However, high-capacity models such as diffusion models can provide more precise estimation as shown in the table above.
> > >
> > > We will include both the GMM and MAF ablation study and this formal Proposition in the revised appendix. We sincerely hope this comprehensive empirical and theoretical analysis fully resolves your remaining concerns.

---

### Official Review · Reviewer_mCGe · 2026-03-10

**Soundness:** 3
**Presentation:** 2
**Significance:** 3
**Originality:** 3
**Overall Recommendation:** 4
**Confidence:** 2

**Summary:**

This paper studies offline cooperative multi-agent RL under multimodal behavior distribution, arguing that standard factorized policy regularization can misalign agents and induce out-of-distribution joint actions. The proposed method OMSD sequentially decomposes the joint behavior policy intro conditional per-agent distributions, learns these conditional with diffusion-based score models, and combines the resulting score regularization with a centralized critic during policy optimization. The paper also presents a stylized proposition illustrating failure of factorized approximations under multimodal joint policies, and reports experiments on a toy bandit task, MPE, and MaMuJoCo benchmarks. The main empirical finding is that OMSD outperforms several offline MARL baselines, with large gains on medium and random-quality datasets.

**Compliance With Llm Reviewing Policy:**

Affirmed.

**Key Questions For Authors:**

1. Can you provide a clearer derivation for Equation 4 to Equation 6, particularly the precise definition of the KL term and the role of updated prefix actions $a_{<i}$?

2. Were the baselines in Table 2 and Table 2 rerun under exactly matched environment versions, observation definitions, and tuning budgets, or were some numbers imported from prior papers?

3. How sensitive is OMSD to agent ordering across more than the one robustness setting alluded to in the paper?

4. Can you provide a quantitative measure showing that OMSD indeed stays closer to dataset support than BPRO-CTDE, MADiff-D, or DoF-P?

**Limitations:**

The author discuss several limitations of their approach in this paper, but a more explicit discussion of generality, computation overhead, and evaluation consistency would improve the transparency of the work.

**Strengths And Weaknesses:**

**Strengths**

The conceptual illustrations are useful. Figure 1 communicates the online-versus-offline multimodality gap clearly, and Figure 2 is especially effective in showing how independent decomposition can create misleading update directions. These figures do actual explanatory work rather than decorative work.

Figure 3 makes a strong case that the authors are not simply using diffusion for its own sake. Treating diffusion models are conditional score estimators, then using those scores as regularizers while keeping decentralized execution, is coherent design.

In Table 2, the improvements on MPE medium/random datasets are substantial. For example, OMSD reaches 137.1 on Predator Prey random, well above the listed baselines. Table 3 also shows very large gains on 6-HalfCHeetah and 3-Hopper. Even if one discounts some of the presentation issues, there is clearly signal here.

**Weaknesses**

The core formal result, Proposition 3.1 is a stylized cautionary example. It does not prove that real offline MARL failures are primarily caused by combinatorial mode shift, nor does it prove that OMSD is unbiased in the general setting. Yet the paper repeatedly speaks in stronger language, including "guarantee coordinated mode selection" and "unbiased reference". This gap matters because the theory currently motivated the ideal, but does not justify the strongest claims.

Equation 4 to 6 are the heart of the method, and they are not presented with sufficient precision. The FL term in Equation 4 is awkwardly defined, the role of $\mu_{-i}$ is opaque, and the transition from reverse-KL BRPO-style reasoning to sequential score regularization is not rigorous enough.

The offline dataset is written as $\mathcal D = (o_t^m, o_t^m, o_{t+1}^m, r_t^m)$, which duplicates the observation and omits the action. Section 2.1 says agents aim to minimize discounted cumulative cost, but the rest of the paper clearly uses rewards and maximization. Page 8 discusses a $\beta$ sweep that does not include 0.3, then cites $\beta =0.3$ as preferred for random datasets.

The whole method conditions on prefix action $a_{<i}$, which means some ordering must be chosen. In homogeneous multi-agent systems, order sensitivity can easily become a hidden inductive bias.

---

> ### Author Rebuttal · Authors · 2026-03-31
>
> We sincerely thank the reviewer for the constructive and detailed feedback, especially for recognizing that our illustrations do "real explanatory work" and that OMSD's design is "coherent."
>
> **W1: On Over-strong Claims**
>
> We agree that terms like "guarantee" and "unbiased" were used too loosely. To clarify, our use of "unbiased" specifically refers to the mathematical property that the chain-rule decomposition $\mu(a \mid s) = \prod_{j=1}^{n} \mu_j(a_j \mid s, a_{\lt j})$ is exact — it introduces no approximation error compared to independent factorization. In the revision, we will replace these terms with more precise descriptions such as "promotes coordinated mode selection" and "provides an exact conditional reference via chain-rule decomposition." We clarify that Proposition 3.1 is intended as a motivational illustration of the CMS pathology, not a general convergence proof. The 2-GMM Bandit experiment and t-SNE visualizations serve as empirical support in complex multimodal scenarios.
>
> **W2/Q1: Precise Derivation of Equations 4 to 6**
>
> We appreciate the request for formal rigor. The transition from a joint KL objective to per-agent sequential score regularization is mathematically exact under coordinate-wise updates.
>
> **1. Joint KL Expansion (Eq. 4).** The objective minimizes the KL divergence between the factorized execution policy $\pi_{\theta}(a \mid s) = \prod_{j=1}^{n} \pi_{\theta_j}(a_j \mid s)$ and the sequentially decomposed behavior policy $\mu(a \mid s) = \prod_{j=1}^{n} \mu_j(a_j \mid s, a_{\lt j})$:
>
> $$D_ {\mathrm{KL}}(\pi_ {\theta} \parallel \mu) = \underset{a \sim \pi_ {\theta}}{\mathbb{E}} \left[ \sum_ {j=1}^{n} \log \pi_ {\theta_j}(a_j \mid s) - \sum_ {j=1}^{n} \log \mu_j(a_j \mid s, a_ {\lt j}) \right]$$
>
> **2. Gradient Separation (Eq. 5).** When computing $\nabla_{\theta_i}$, all terms $j \neq i$ vanish: $\log \pi_{\theta_j}$ terms have decoupled parameters, and the prefix actions $a_{\lt i}$ sampled from $\pi_{\lt i}$ are treated as fixed constants (gradient flow is disconnected at the sampling operation). Therefore:
>
> $$\nabla_ {\theta_i}\mathcal{L}^i = \underset{s, \, a_ {\lt i} \sim \pi_ {\lt i}}{\mathbb{E}} \left[ \left( \nabla_ {a_i} Q^{\mathrm{tot}}(s,a) + \frac{1}{\beta}\nabla_ {a_i}\log \mu_i(a_i \mid s, a_ {\lt i}) \right) \nabla_ {\theta_i}\pi_ {\theta_i} \right]$$
>
> This is mathematically exact, ensuring each agent is regularized against the correct conditional mode of the dataset.
>
> **3. Score Estimation (Eq. 6).** The term $\nabla_{a_i}\log \mu_i(a_i \mid s, a_{\lt i})$ is the Stein score of the conditional behavior distribution. We approximate it via the pretrained diffusion model's denoising output:
>
> $$\nabla_ {a_i} \log \mu_i(a_i \mid s, a_ {\lt i}) \approx -\hat{\epsilon}_ {i}(a_i, s, a_ {\lt i}, t) / \sigma_t$$
>
> **4. Role of $\mu_{-i}$.** We will remove this redundant notation in the revision and directly present the sequential per-agent derivation above.
>
> **W3: Notation and Consistency Errors**
>
> We will correct $\mathcal{D}$ to include actions: $\mathcal{D} = (o_t, a_t, o_{t+1}, r_t)$. We will fix the optimization objective to reward maximization throughout, removing inconsistent "cost minimization" in Sec. 2.1. The reported $\beta=0.3$ is a typo — the actual sweep includes $\beta=0.5$, which is the preferred value consistent with our Appendix experimental settings.
>
> **W4/Q3: Agent Ordering Sensitivity**
>
> Sequential conditioning is a necessary design to resolve mode ambiguity from multiple equilibria. Crucially, this asymmetry exists only during the training phase's gradient computation. During execution, each agent's learned policy $\pi_{\theta_i}(a_i \mid s)$ is fully independent — it takes only the state $s$ as input, with no dependence on $a_{\lt i}$. All agents execute in a fully decentralized, parallel, and symmetric manner. The ablation in Fig.10 with multiple randomized update orders shows highly consistent convergence, demonstrating practical permutation invariance. The t-SNE visualizations in Fig.9 further confirm the ordering serves as a coordination mechanism rather than a mode bias. We also provide a direct GMM vs. Diffusion ablation (see our response to Reviewer jcJo) that further validates the framework's robustness.
>
> **Q2: Baseline Sources and Consistency**
>
> Baseline scores were imported from their respective original publications. To ensure comparability, we strictly adopted the same dataset formats, environment versions, and evaluation protocols as each original paper. We will explicitly label the source of each baseline in the revised Appendix.
>
> **Q4: Quantitative OOD Measure**
>
> Our t-SNE visualizations (Fig. 4c, Fig. 9) qualitatively show that OMSD's policies remain within high-density dataset regions, a direct consequence of sequential score regularization constraining each agent toward in-distribution modes. We will include quantitative OOD metrics (e.g., action log-likelihood under behavior policy) in the revised appendix.

---

### Official Review · Reviewer_8rze · 2026-03-10

**Soundness:** 3
**Presentation:** 3
**Significance:** 2
**Originality:** 2
**Overall Recommendation:** 4
**Confidence:** 3

**Summary:**

This paper studies offline multi-agent reinforcement learning (MARL) under multimodal behavior distributions. The authors argue that many existing offline MARL methods rely on independent policy regularization, which may cause distribution shift and mis-coordination among agents when the dataset contains multiple joint behavior modes.
To address this issue, the paper proposes **OMSD**, which models the behavior policy using a sequential conditional decomposition across agents. Diffusion models are used to estimate score functions of the conditional behavior distributions, which serve as regularization gradients during policy optimization together with a centralized critic.
Experiments on MPE and MaMuJoCo benchmarks show strong improvements over existing baselines, especially on medium and random datasets where multimodal behaviors are more prominent.

**Compliance With Llm Reviewing Policy:**

Affirmed.

**Final Justification:**

The authors' rebuttal have addressed my main concern. So I decide to finalize my recommendation to 4 from 3.

**Key Questions For Authors:**

1. Although the paper does not include MADiff-C as a baseline, checking the results reported in the MADiff paper suggests that even the centralized version (MADiff-C) performs much worse than OMSD on the MPE Random datasets. Since MADiff-C models joint actions centrally, should it theoretically avoid the distribution shift caused by independent policy factorization discussed in this paper? Why does OMSD still achieve significantly better performance than MADiff-C in such settings?

2. The policy update combines the diffusion-based behavior score with gradients from a centralized critic. How sensitive is the method to critic estimation errors, particularly in settings with low-quality datasets such as random datasets?

**Limitations:**

yes

**Strengths And Weaknesses:**

**Strengths**:

1. The paper highlights the issue of multimodal joint behavior distributions in offline MARL and explains why independent policy regularization may fail. It provides clear motivation and formulation.

2. The experimental results are impressive, showing consistent improvements over strong baselines across multiple benchmarks, especially on medium and random datasets.

3. The paper uses diffusion models to estimate score functions of conditional behavior distributions, which serve as regularization gradients during policy updates. This approach avoids expensive sampling from diffusion policies while still leveraging the expressive power of diffusion-based behavior modeling.

**Weaknesses**:

1. While effective, the core idea of sequential policy modeling is not entirely new. Similar mechanisms have been explored in online MARL algorithms (e.g., sequential policy updates or HAPPO-style methods) and in autoregressive policy modeling. Therefore, the novelty mainly lies in applying this idea to offline MARL with diffusion-based regularization.

2. The proposed decomposition introduces a fixed ordering among agents. Although the paper suggests that the method is not very sensitive to ordering, this design still breaks agent symmetry and may affect robustness in environments where agents are homogeneous.

3. Although the paper includes two diffusion-model-based offline MARL methods, MADiff-D and DoF-P, it does not include a comparison with MADiff-C. Adding this comparison would make the experimental analysis more comprehensive.

---

> ### Author Rebuttal · Authors · 2026-03-31
>
> We sincerely thank the reviewer for the comprehensive summary and for recognizing the significant empirical improvements our method brings. We are also encouraged that you share the consensus with other reviewers (e.g., Reviewer qh4B, mCGe) that identifying the CMS problem is a critical and well-reasoned contribution to offline MARL.
>
> **W1: On the Novelty of Sequential Policy Decomposition**
>
> We agree that sequential/autoregressive modeling is an established paradigm in online MARL (e.g., HAPPO, MAT). However, our core contribution is not the sequential form itself, but identifying that independent regularization leads to CMS under multimodal data, and demonstrating via Proposition 3.1 and empirical evidence that this causes severe distribution shift. As Reviewer qh4B noted, OMSD is "the first practical framework specifically designed" to tackle this offline challenge. We will add a dedicated discussion on sequential modeling in MARL to clarify our contributions.
>
> Introducing sequential conditional decomposition within BRPO is not meant to change the execution paradigm or guarantee monotonic improvement. Rather, it is mathematically the most concise, direct, and effective way to achieve coordinated individual regularization. We leverage diffusion models because only high-capacity estimators can accurately capture complex multimodal distributions (see our GMM ablation in response to Reviewer jcJo). Our contribution lies in seamlessly embedding this insight into offline policy regularization, achieving powerful mode-seeking capabilities.
>
> **W2: On Breaking Agent Symmetry**
>
> We fully understand the concern that a fixed ordering might affect robustness in environments where agents are homogeneous. However, in the offline multimodal setting, breaking symmetry during training is not a flaw but a necessary design to resolve the ambiguity of multiple equilibria. Although homogeneous agents share identical capabilities, the multimodal behavior distribution still requires coordination to resolve mode ambiguity. By introducing prefix actions as conditions, we lower the difficulty of modeling each agent's behavior distribution and ensure agents align with the same joint mode during individual updates, without embedding any agent-specific bias.
>
> Crucially, this asymmetry exists strictly during the training phase's gradient computation. During execution, each agent's policy $\pi_{\theta_i}(a_i \mid s)$ takes only state $s$ as input with no dependence on prefix actions — all agents remain fully decentralized, parallel, and symmetric. Our ablation in Fig. 10 with multiple randomized update orders shows highly consistent convergence, demonstrating practical permutation invariance. The t-SNE visualizations in Fig. 9 further confirm that OMSD captures diverse viable modes rather than collapsing to a single one.
>
> **W3 & Q1: Comparison with MADiff-C on Random Datasets**
>
> MADiff-C avoids CMS by design through centralized joint action modeling. However, it introduces a different and more severe form of distribution shift — cascading sampling errors inherent to generation-based methods. As acknowledged in the MADiff paper's Limitations (Sec 5.6), this is particularly damaging on low-quality datasets where the generative model must produce entire action sequences from noisy demonstrations. In contrast, OMSD is a policy-based method that combines behavior modeling and value estimation. By leveraging IQL's conservative value estimation to extract reliable value signals even from noisy data, combined with coordinated score regularization that constrains each agent toward in-distribution modes, OMSD is fundamentally more robust to suboptimal datasets. We appreciate the suggestion and will include a dedicated discussion about MADiff-C in our revised paper.
>
> **Q2: Sensitivity to Critic Estimation Errors**
>
> OMSD mitigates critic errors through complementary safeguards: (1) IQL avoids overestimation by strictly evaluating in-sample actions, serving as a reliable value extractor even under poor data; (2) coordinated score regularization confines exploration near the behavior policy's support set, preventing OOD divergence even when the critic is imprecise.
>
> Empirically, the $\beta$ sweep in Fig. 4b directly addresses this: $\beta$ controls the relative weight between critic gradient and score regularization. OMSD maintains stable performance across a wide range of $\beta \in \{0.001, ..., 0.5\}$, indicating robustness to this balance. If the method were highly sensitive to critic errors, we would expect sharp degradation as $\beta$ increases (amplifying critic influence), which is not observed. This, combined with strong Random dataset performance where critic quality is inherently lower, confirms OMSD's practical robustness.

---

> > ### Author Rebuttal · Reviewer_8rze · 2026-04-03
> >
> > Thanks for the authors' rebuttal. I have raised my score.

---

> > > ### Author Response · Authors · 2026-04-05
> > >
> > > Dear Reviewer 8rze,
> > >
> > > We sincerely thank you for your active participation in the discussion and for acknowledging our efforts. We are glad that our rebuttal and the additional discussions have successfully addressed your concerns. Your constructive feedback has been instrumental in improving the quality and clarity of our manuscript.
> > >
> > > Best regards,
> > >
> > > The Authors

---

### Official Review · Reviewer_qh4B · 2026-03-16

**Soundness:** 3
**Presentation:** 3
**Significance:** 3
**Originality:** 3
**Overall Recommendation:** 4
**Confidence:** 3

**Summary:**

This manuscript targets the challenge of multimodal joint behavior policies in offline cooperative multi-agent reinforcement learning (MARL), which often leads to coordination failure and distribution shift. It claims that existing methods relying on independent policy factorization struggle to handle the complex inter-agent dependencies present in datasets collected from diverse sources. To fill in this gap, the authors propose OMSD, a novel framework that leverages a sequential decomposition of the joint behavior policy. OMSD employs diffusion-based generative models to estimate conditional score functions for each agent, providing coordinated regularization during policy updates. Moreover, OMSD is validated on a bandit example and multiple challenging continuous control benchmarks (MPE and MaMuJoCo), consistently outperforming state-of-the-art methods, especially on datasets with high behavioral diversity.

**Compliance With Llm Reviewing Policy:**

Affirmed.

**Final Justification:**

From my point of view, I think the authors have addressed nearly all of my concerns. Thus, I intend to raise my original score.

**Key Questions For Authors:**

Please refer to the weakness part.

**Limitations:**

Yes.

**Strengths And Weaknesses:**

Strength:

1.	OMSD is the first practical framework specifically designed to tackle the multimodal joint behavior policy problem in offline MARL by introducing a sequential decomposition mechanism, filling a critical research gap in the field.

2.	OMSD uses Sequential Score Decomposition to provide coordinated regularization for each agent, effectively addressing the "Combinatorial Mode Shift" (CMS) problem as illustrated in Figure 2, thereby avoiding out-of-distribution (OOD) joint actions and improving policy coordination.

3.	OMSD is validated on multiple public datasets (MPE and MaMuJoCo) from different sources (OMAR, OG-MARL, OMIGA) and considers various data qualities (expert, medium, random), exhibiting robust and promising performance.

Weakness:

1.	The method relies on diffusion models and DiLac policies, which are designed for continuous actions. The paper explicitly states (Appendix J) that discrete action spaces are not supported, limiting applicability to discrete MARL domains (e.g., SMAC, Google Research Football).

2.	Pretraining separate conditional diffusion models for each agent (plus a joint critic) requires significant computation (6-10 hours for IQL + 4-6 hours per agent for diffusion on RTX 3090). The scalability to large teams (>10 agents) is not demonstrated.

3.	The experimental results in Tables 2 and 3 are reported with mean and standard error over 5 seeds, which is a relatively small number of runs. Providing standard deviations or confidence intervals would offer a more robust measure of statistical significance and result stability.

---

> ### Author Rebuttal · Authors · 2026-03-31
>
> We sincerely thank the reviewer for the positive assessment and constructive feedback, especially for recognizing that OMSD "fills a critical research gap," "effectively addresses the CMS problem," and exhibits "robust and promising performance" across diverse datasets. Regarding the concerns:
>
> **W1: On the Limitation to Continuous Action Spaces.**
>
> As acknowledged in Appendix J, OMSD's current implementation is indeed tailored for continuous tasks. However, its core insights about CMS problems and coordinated regularization are universally applicable in both continuous and discrete tasks. Discrete MARL also faces similar multimodality and OOD joint action challenges (e.g., OMAR, OMIGA, AlberDICE). Adapting to discrete spaces would require discrete generative modeling (e.g., discrete diffusion), but the coordinated regularization framework remains directly applicable.
>
> We prioritized continuous tasks because precise multimodal regularization is inherently more challenging in continuous spaces. Traditional unimodal policies (e.g., Gaussians) fail severely under multimodal distributions, introducing biased behavior regularization that heavily hinders the application of policy-based offline MARL methods. This specific challenge strictly necessitates the introduction of high-capacity generative models like diffusion. Extending OMSD to discrete domains is a promising future direction beyond our current scope, and we will explicitly clarify this design motivation in the revision.
>
> **W2: On the Limitation to Large Teams**
>
> We acknowledge that OMSD requires additional computational resources for behavior policy training compared to critic-only methods. However, a key architectural advantage of OMSD is that the conditional diffusion models are fully decoupled during the pretraining phase. Consequently, this stage is fully parallelizable across agents, making the diffusion phase flexible and scalable to larger teams.
>
> We have empirically demonstrated this scalability on the 6-agent HalfCheetah task, which belongs to the larger end of existing offline MARL benchmarks in continuous control tasks. To ensure fair benchmarking, our evaluation strictly followed standard continuous control tasks from existing offline MARL literature, which currently lacks established datasets and baselines for larger-scale scenarios (e.g., >10 agents). We will explicitly highlight this parallelization discussion in the revision and leave larger team benchmarks to future work.
>
> **W3: On Statistical Reporting**
>
> We agree with the reviewer that std deviation (SD) provides a more robust measure of performance. We originally reported std errors (SE) strictly to align with the reported metrics in the baseline literature for a direct numerical comparison in Table 2. We have updated Table 2 in the revised manuscript to report the mean and SD across all 5 random seeds (calculated as $SD = SE \times \sqrt{5}$). Due to rebuttal character limits, we provide a condensed version of the updated Table 2 below, focusing on the challenging Random datasets across MPE tasks against the baselines. As demonstrated, OMSD maintains a stable and statistically significant performance gain.
>
> | **MPE Task (Random)** | **MADiff-D** | **DoF-P** | **OMSD (SD) [Ours]** |
> | --------------------- | ----------------- | -------------- | -------------------- |
> | Cooperative Nav.      | 6.9 ± 6.9         | 34.5 ± 12.1    | **69.8 ± 10.3**      |
> | Predator Prey         | 3.2 ± 8.9         | 14.8 ± 7.2     | **133.9 ± 16.5**     |
> | World                 | 2.0 ± 6.7         | 15.1 ± 6.7     | **141.1 ± 13.0**     |

---

> > ### Author Rebuttal · Reviewer_qh4B · 2026-04-07
> >
> > From my point of view, I think the authors have addressed nearly all of my concerns. Thus, I intend to raise my original score.

---

> > > ### Author Response · Authors · 2026-04-07
> > >
> > > Dear Reviewer qh4B,
> > >
> > > We sincerely thank you for your active participation in the discussion and for acknowledging our efforts. We are glad that our rebuttal and the additional discussions have successfully addressed your concerns. Your constructive feedback has been instrumental in improving the quality and clarity of our manuscript.
> > >
> > > Best regards,
> > >
> > > The Authors

---

### Decision · Program_Chairs · 2026-04-30

**Decision:**

Accept (regular)

**Comment:**

I took a look at the un-responded rebuttal points and believe the authors have adequately addressed the main concerns of the two reviewers. The paper is of sufficient quality to clear the bar for ICML.